# Pre-trained Perceptual Features
# Improve Differentially Private Image Generation

**Frederik Harder**                                                            *fharder@tue.mpg.de*
*Max Planck Institute for Intelligent Systems & University of Tübingen*

**Milad Jalali Asadabadi**                                                     *miladj7@cs.ubc.ca*
*University of British Columbia*

**Danica J. Sutherland**                                                       *dsuth@cs.ubc.ca*
*University of British Columbia & Alberta Machine Intelligence Institute*

**Mijung Park**                                                                *mijungp@cs.ubc.ca*
*University of British Columbia & Alberta Machine Intelligence Institute*

**Reviewed on OpenReview:** *https://openreview.net/forum?id=R6W7zkMz0P*

## Abstract

Training even moderately-sized generative models with differentially-private stochastic gradient descent (DP-SGD) is difficult: the required level of noise for reasonable levels of privacy is simply too large. We advocate instead building off a good, relevant representation on an informative public dataset, then learning to model the private data with that representation. In particular, we minimize the maximum mean discrepancy (MMD) between private target data and a generator's distribution, using a kernel based on perceptual features learned from a public dataset. With the MMD, we can simply privatize the data-dependent term once and for all, rather than introducing noise at each step of optimization as in DP-SGD. Our algorithm allows us to generate CIFAR10-level images with $\epsilon \approx 2$ which capture distinctive features in the distribution, far surpassing the current state of the art, which mostly focuses on datasets such as MNIST and FashionMNIST at a large $\epsilon \approx 10$. Our work introduces simple yet powerful foundations for reducing the gap between private and non-private deep generative models. Our code is available at `https://github.com/ParkLabML/DP-MEPF`.[1]

## 1 INTRODUCTION

The gold standard privacy notion, *differential privacy* (DP), is now ubiquitous in a diverse range of academic research, industry products (Apple, 2017), and even government databases (National Conference of State Legislatures, 2021). DP provides a mathematically provable privacy guarantee, which is its main strength and reason for its popularity. DP even offers means of tracking the effect of multiple accesses to the same data on it's overall privacy level, but with each access, the privacy of the data gradually degrades. To guarantee a high level of privacy, one thus needs to limit access to data, a challenge in applying DP with the usual iterative optimization algorithms used in machine learning.

Differentially private data generation solves this problem by creating a synthetic dataset that is *similar* to the private dataset, in terms of some chosen similarity metric. While producing such a synthetic dataset incurs a privacy loss, the resulting dataset can be used repeatedly without further loss of privacy. Classical approaches, however, typically assume a certain class of pre-specified purposes on how the synthetic data can be used (Mohammed et al., 2011; Xiao et al., 2010; Hardt et al., 2012; Zhu et al., 2017). If data analysts use the data for other tasks outside these pre-specified purposes, the theoretical guarantees on its utility are lost.

---

[1]This is a revision of the first published version which contained erroneous FID scores. Please refer to this paper's OpenReview page for a clarification of our errors and the older version.

To produce synthetic data usable for potentially *any* purpose, many papers on DP data generation have utilized the recent advances in deep generative modelling. The majority of these approaches are based on the generative adversarial network (GAN; Goodfellow et al., 2014) framework, where a discriminator and a generator play an adversarial game to optimize a given distance metric between the true and synthetic data distributions. Most approaches under this framework have used DP-SGD (Abadi et al., 2016), where the gradients of the discriminator (which compares generated samples to private data) are privatized in each training step, resulting in a high overall privacy loss (Park et al., 2017; Torkzadehmahani et al., 2019; Yoon et al., 2019; Xie et al., 2018; Frigerio et al., 2019). Another challenge is that, as the gradients must have bounded norm to derive the DP guarantee, the amount of noise for privatization in DP-SGD increases proportionally to the dimension of the discriminator. Hence, these methods are typically bound to relatively small discriminators, limiting the ability to learn data distributions beyond, say, MNIST (LeCun & Cortes, 2010) or FashionMNIST (Xiao et al., 2017).

Given these challenges, the heavy machinery such as GANs and large-scale auto-encoder-based methods – capable of generating complex datasets in a non-private setting – fails to model datasets such as CIFAR-10 (Krizhevsky, 2009) or CelebA (Liu et al., 2015) with a meaningful privacy guarantee (e.g., $\epsilon \approx 2$). Typical deep generative modeling papers have moved well beyond these datasets, but to the best of our knowledge, currently there is no DP data generation method that can produce reliable samples at a reasonable privacy level.

How can we reduce this huge gap between the performance of non-private deep generative models and that of private counterparts? We argue that we can narrow this gap by using the abundant resource of *public* data, in line with the core message of Tramèr & Boneh (2021): *We simply need better features for differentially private learning.* While Tramèr & Boneh demonstrated this in the context of DP classification, we aim to show the applicability of this reasoning for the more challenging problem of DP data generation, with a focus on high-dimensional image generation.

We propose to exploit public data to learn *perceptual features* (PFs) from public data, which we will use to compare synthetic and real data distributions. Following dos Santos et al. (2019), we use "perceptual features" to mean the vector of all activations of a pretrained deep network for a given data point, e.g. the hundreds of thousands of hidden activations from applying a trained deep classifier to an image. Building on dos Santos et al. (2019), who use PFs for transfer learning in natural image generation, our goal is to improve the quality of natural images generated with differential privacy constraints.

We construct a kernel on images using these powerful PFs, then train a generator by minimizing the Maximum Mean Discrepancy (MMD) (Gretton et al., 2012) between distributions (as in Harder et al., 2021; Li et al., 2015; Dziugaite et al., 2015; dos Santos et al., 2019). This scheme is non-adversarial, leading to simpler and more stable optimization; moreover, it allows us to privatize the mean embedding of the private dataset *once*, using it at each step of generator training without incurring cumulative privacy losses.

We observe in our experiments that as long as the public data contains more complex patterns than private data, e.g., transferring the knowledge learned from ImageNet as public data to generate CIFAR-10 images as private data, the learned features from public data are useful enough to generate good synthetic data. We successfully generate reasonable samples for CIFAR-10, CelebA, MNIST, and FashionMNIST in high-privacy regimes. We also theoretically analyze the effect of privatizing our loss function, helping understand the privacy-accuracy trade-offs in our method.

The main point of our paper is that features from public data are a key tool for improved DP data generation, a point we think our experiments make resoundingly; this may be "obvious", but has not been explored for image generation. Our proposed method, in particular, is a simple (which, we think, is a good thing) initial technique exploiting this idea, which outperforms simple pretraining of DP-GAN and DP-Sinkhorn (see Section 6). We hope this work will inspire future work on other ways to use public features for improving image generation with differential privacy.

## 2  BACKGROUND

We provide background information on maximum mean discrepancy and differential privacy.

**Maximum Mean Discrepancy**   The MMD is a distance between distributions based on a kernel $k_\phi(x, y) = \langle \phi(x), \phi(y) \rangle_{\mathcal{H}}$, where $\phi$ maps data in $\mathcal{X}$ to a Hilbert space $\mathcal{H}$ (Gretton et al., 2012). One definition is

$$\text{MMD}_{k_\phi}(P, Q) = \big\| \mathbb{E}_{x \sim P}[\phi(x)] - \mathbb{E}_{y \sim Q}[\phi(y)] \big\|_{\mathcal{H}},$$

where $\boldsymbol{\mu}_\phi(P) = \mathbb{E}_{x \sim P}[\phi(x)] \in \mathcal{H}$ is known as the (kernel) *mean embedding* of $P$, and is guaranteed to exist if $\mathbb{E}_{x \sim P}\sqrt{k(x, x)} < \infty$ (Smola et al., 2007). If $k_\phi$ is *characteristic* (Sriperumbudur et al., 2011), then $P \mapsto \boldsymbol{\mu}_\phi(P)$ is injective, so $\mathrm{MMD}_{k_\phi}(P, Q) = 0$ if and only if $P = Q$.

For a sample set $\mathcal{D} = \{\mathbf{x}_i\}_{i=1}^m \sim P^m$, the empirical mean embedding $\boldsymbol{\mu}_\phi(\mathcal{D}) = \frac{1}{m}\sum_{i=1}^m \phi(\mathbf{x}_i)$ is the "plug-in" estimator of $\boldsymbol{\mu}_\phi(P)$ using the empirical distribution of $\mathcal{D}$. Given $\tilde{\mathcal{D}} = \{\tilde{\mathbf{x}}_i\}_{i=1}^n \sim Q^n$, we can estimate $\mathrm{MMD}_{k_\phi}(P, Q)$ as the distance between empirical mean embeddings,

$$\mathrm{MMD}_{k_\phi}(\mathcal{D}, \tilde{\mathcal{D}}) = \left\| \frac{1}{m}\sum_{i=1}^m \phi(\mathbf{x}_i) - \frac{1}{n}\sum_{i=1}^n \phi(\tilde{\mathbf{x}}_i) \right\|_{\mathcal{H}}. \tag{1}$$

We would like to minimize the distance between a target data distribution $P$ (based on samples $\mathcal{D}$) and the output distribution $Q_{g_\theta}$ of a generator network $g_\theta$. If the feature map is finite-dimensional and norm-bounded, following Harder et al. (2021); Vinaroz et al. (2022), we can privatize the mean embedding of the data distribution $\boldsymbol{\mu}_\phi(\mathcal{D})$ with a known DP mechanism such as the Gaussian or Laplace mechanisms, to be discussed shortly. As the summary of the real data does not change over the course of a generator training, we only need to privatize $\boldsymbol{\mu}_\phi(\mathcal{D})$ once.

**Differential privacy**   A mechanism $\mathcal{M}$ is $(\epsilon, \delta)$-DP for a given $\epsilon \geq 0$ and $\delta \geq 0$ if and only if

$$\Pr[\mathcal{M}(\mathcal{D}) \in S] \leq e^\epsilon \cdot \Pr[\mathcal{M}(\mathcal{D}') \in S] + \delta$$

for all possible sets of the mechanism's outputs $S$ and all neighbouring datasets $\mathcal{D}, \mathcal{D}'$ that differ by a single entry. One of the most well-known and widely used DP mechanisms is the *Gaussian mechanism*. The Gaussian mechanism adds a calibrated level of noise to a function $\mu : \mathcal{D} \mapsto \mathbb{R}^p$ to ensure that the output of the mechanism is $(\epsilon, \delta)$-DP: $\widetilde{\mu}(\mathcal{D}) = \mu(\mathcal{D}) + n$, where $n \sim \mathcal{N}(0, \sigma^2 \Delta_\mu^2 \mathbf{I}_p)$. Here, $\sigma$ is often called a privacy parameter, which is a function[2] of $\epsilon$ and $\delta$. $\Delta_\mu$ is often called the *global sensitivity* (Dwork et al., 2006), which is the maximum difference in $L_2$-norm given two neighbouring $\mathcal{D}$ and $\mathcal{D}'$, $||\mu(\mathcal{D}) - \mu(\mathcal{D}')||_2$. In this paper, we will use the Gaussian mechanism to ensure the mean embedding of the data distribution is DP.

## 3   METHOD

In this paper, to transfer knowledge from public to private data distributions, we construct a particular kernel $k_\Phi$ to use in Equation 1 based on *perceptual features* (PFs).

### 3.1   MMD with perceptual features as a feature map

We call our proposed method *Differentially Private Mean Embeddings with Perceptual Features (DP-MEPF)*, analogous to the related method DP-ME**R**F (Harder et al., 2021). We use high-dimensional, over-complete perceptual features from a feature extractor network pre-trained on a public dataset, as illustrated in **Step 1** of Figure 1. Given a vector input $\mathbf{x}$, the pre-trained feature extractor network outputs the perceptual features from each layer, where the $j$th layer's PF is denoted by $\mathbf{e}_j(\mathbf{x})$. Each of the $J$ layers' perceptual features is of a different length, $\mathbf{e}_j(\mathbf{x}) \in \mathbb{R}^{d_j}$; the total dimension of the perceptual feature vector is $D = \sum_{j=1}^J d_j$.

As illustrated in **Step 2** in Figure 1, we use those PFs to form our feature map $\Phi(\mathbf{x}) := [\boldsymbol{\phi}_1(\mathbf{x}), \boldsymbol{\phi}_2(\mathbf{x})]$, where the first part comes from a concatenation of PFs from all the layers: $\boldsymbol{\phi}_1(\mathbf{x}) = [\mathbf{e}_1(\mathbf{x}), \cdots, \mathbf{e}_J(\mathbf{x})]$, while the second part comes from their squared values: $\boldsymbol{\phi}_2(\mathbf{x}) = [\mathbf{e}_1^2(\mathbf{x}), \cdots, \mathbf{e}_J^2(\mathbf{x})]$, where $\mathbf{e}_j^2(\mathbf{x})$ means each entry of $\mathbf{e}_j(\mathbf{x})$ is squared. Using this feature map, we then construct the mean embedding of a data distribution given the data samples $\mathcal{D} = \{\mathbf{x}_i\}_{i=1}^m$:

$$\boldsymbol{\mu}_P(\mathcal{D}) = \begin{bmatrix} \boldsymbol{\mu}_P^{\phi_1}(\mathcal{D}) \\ \boldsymbol{\mu}_P^{\phi_2}(\mathcal{D}) \end{bmatrix} = \begin{bmatrix} \frac{1}{m}\sum_{i=1}^m \boldsymbol{\phi}_1(\mathbf{x}_i) \\ \frac{1}{m}\sum_{i=1}^m \boldsymbol{\phi}_2(\mathbf{x}_i) \end{bmatrix}. \tag{2}$$

Lastly (**Step 3** in Figure 1), we will train a generator $g_\theta$ that maps latent vectors $\mathbf{z}_i \sim \mathcal{N}(0, I)$ to a synthetic data sample $\tilde{\mathbf{x}}_i = g_\theta(\mathbf{z}_i)$; we need to find good parameters $\theta$ for the generator. In non-private settings, we estimate the generator's

---

[2]The relationship can be numerically computed by packages like `auto-dp` (Wang et al., 2019), among other methods.

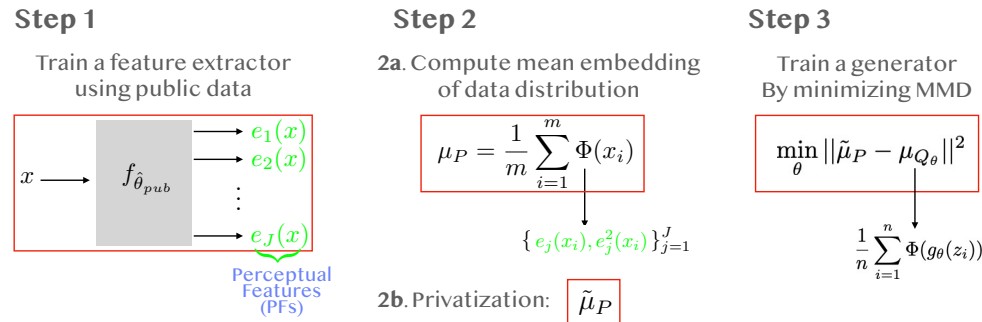

Figure 1: Three steps in *differentially private mean embedding with perceptual features (DP-MEPF)*. **Step 1:** We train a feature extractor neural network, $f_{\hat{\theta}_{pub}}$, using public data. This is a function of public data, with no privacy cost to train. A trained $f_{\hat{\theta}_{pub}}$ maps an input $\mathbf{x}$ to perceptual features (in green), the outputs of each layer. **Step 2:** We compute the mean embedding of the data distributions using a feature map consisting of the first and second moments (in green) of the perceptual features, and privatize it based on the Gaussian mechanism (see text). **Step 3:** We train a generator $g_{\theta}$, which produces synthetic data from latent codes $z_i \sim \mathcal{N}(0, I)$, by minimizing the privatized MMD.

parameters by minimizing an estimate of $\mathrm{MMD}^2_{k_\Phi}(P, Q_{g_\theta})$, using $\tilde{\mathcal{D}} = \{\tilde{\mathbf{x}}_i\}$ in Equation 1, similar to Dziugaite et al. (2015); Li et al. (2015); dos Santos et al. (2019). In private settings, we privatize $\mathcal{D}$'s mean embedding to $\tilde{\boldsymbol{\mu}}_\phi(\mathcal{D})$ with the Gaussian mechanism (details below), and minimize

$$\widetilde{\mathrm{MMD}}^2_{k_\Phi}(\mathcal{D}, \tilde{\mathcal{D}}) = \left\| \tilde{\boldsymbol{\mu}}_\phi(\mathcal{D}) - \boldsymbol{\mu}_\phi(\tilde{\mathcal{D}}) \right\|^2. \tag{3}$$

A natural question that arises is whether the MMD using the PFs is a metric: if $\mathrm{MMD}_{k_\Phi}(P, Q) = 0$ only if $P = Q$. As PFs have a finite-dimensional embedding, we in fact know this cannot be the case (Sriperumbudur et al., 2011). Thus, there exists *some* pair of distributions which our MMD cannot distinguish. However, given that linear functions in perceptual feature spaces can obtain excellent performance on nearly any natural image task (as observed in transfer learning), it seems that PFs are "nearly" universal for natural distributions of images (dos Santos et al., 2019). Thus we expect the MMD with this kernel to do a good job of distinguishing "natural" distributions from one another, though the possibility of "adversarial attacks" perhaps remains.

A more important question in our context is whether this MMD serves as a good loss for training a generator, and whether the resulting synthetic data samples are reasonably faithful to the original data samples. Our experiments in Section 6, as well as earlier work by dos Santos et al. (2019) in non-private settings, imply that it is.

**Privatization of mean embedding** We privatize the mean embedding of the data distribution only once, and reuse it repeatedly during the training of the generator $g_{\boldsymbol{\theta}}$. We use the Gaussian mechanism to separately privatize the first and second parts of the feature map. We normalize each type of perceptual features such that $\|\boldsymbol{\phi}_1(\mathbf{x}_i)\|_2 = 1$ and $\|\boldsymbol{\phi}_2(\mathbf{x}_i)\|_2 = 1$ for each sample $\mathbf{x}_i$. After this change, the sensitivity of each part of the mean embedding is

$$\max_{\mathcal{D}, \mathcal{D}' \text{ s.t. } |\mathcal{D} - \mathcal{D}'| = 1} \|\boldsymbol{\mu}_{\boldsymbol{\phi}_t}(\mathcal{D}) - \boldsymbol{\mu}_{\boldsymbol{\phi}_t}(\mathcal{D}')\|_2 \le \frac{2}{m}, \tag{4}$$

where $\boldsymbol{\mu}_{\boldsymbol{\phi}_t}(\mathcal{D})$ denotes the two parts of the mean embedding for $t = 1, 2$. Using these sensitivities, we add Gaussian noise to each part of the mean embedding, obtaining

$$\tilde{\boldsymbol{\mu}}_\Phi(\mathcal{D}) = \begin{bmatrix} \tilde{\boldsymbol{\mu}}_{\boldsymbol{\phi}_1}(\mathcal{D}) \\ \tilde{\boldsymbol{\mu}}_{\boldsymbol{\phi}_2}(\mathcal{D}) \end{bmatrix} = \begin{bmatrix} \frac{1}{m} \sum_{i=1}^m \boldsymbol{\phi}_1(\mathbf{x}_i) + \mathbf{n}_1 \\ \frac{1}{m} \sum_{i=1}^m \boldsymbol{\phi}_2(\mathbf{x}_i) + \mathbf{n}_2 \end{bmatrix}, \tag{5}$$

where $\mathbf{n}_t \sim \mathcal{N}(0, \frac{4\sigma^2}{m^2} I)$ for $t = 1, 2$.

Since we are using the Gaussian mechanism twice, we simply compose the privacy losses from each mechanism. More precisely, given a desired privacy level $\epsilon, \delta$, we use the package of Wang et al. (2019) to find the corresponding $\sigma$ for the two Gaussian mechanisms.

**Labeled data generation** Extending our framework to generate both labels and input images is straightforward. As done by Harder et al. (2021), we construct a separate mean embedding for each class-conditional input distribution and then concatenate them into a single embedding

$$\tilde{\boldsymbol{\mu}}_{\boldsymbol{\phi}_t}(\mathcal{D}) = \left[\frac{1}{m}\sum_{i\in C_1}\boldsymbol{\phi}_t(\mathbf{x}_i) + \mathbf{n}_{t,1} \quad \cdots \quad \frac{1}{m}\sum_{i\in C_K}\boldsymbol{\phi}_t(\mathbf{x}_i) + \mathbf{n}_{t,K}\right]^\top,\tag{6}$$

where $K$ is the number of classes and $C_k = \{i \in [m] | y_i = k\}$ is the set of indices belonging to class $k$. As a result, the size of the final mean embedding is $D \times K$ (number of perceptual features by the number of classes) if we use only the first moment, or $2 \times D \times K$ if we use the first two moments. This is exactly the conditional mean embedding with a discrete kernel on the class label (Song et al., 2013). In the case of imbalanced data, an estimate of the label distribution can be obtained at low privacy cost with a DP release of the class counts, as done in Harder et al. (2021). Since all datasets considered in this paper are balanced, this step is not necessary in our experiments.

## 3.2 Differentially private early stopping

On some datasets (CelebA and Cifar10) we observe that the generated sample quality deteriorates if the model is trained for too many iterations in high-privacy settings. This is indicated by a steady increase in FID score (Heusel et al., 2017), and likely due to overfitting to the static noisy embedding. Since the FID score is based on the training data, simply choosing the iteration with the best FID score after training has completed would violate privacy.

Privatizing the FID score requires privatizing the covariance of the output of the final pooling layer in the Inception network, which is quite sensitive. Instead, we privatize the first and second moment of data embeddings as in Equation 2, but using only the output of the final pooling layer in the Inception network. We then use this quantity as a private proxy for FID, and select the iteration with the lowest score. To minimize the privacy cost, we choose a larger noise parameter than for the main objective: $\sigma_{stopping} = 10\sigma$, where $\sigma$ is the noise scale for privatizing each part of the data mean embeddings, works well. Again, we compose these $\sigma$s with the analysis of Wang et al. (2019).

# 4 THEORETICAL ANALYSIS

We now bound the effect of adding noise to our loss function, showing that asymptotically our noise does not hurt the rate at which our model converges to the optimal model.

Appendix A proves full finite-sample versions of all of the following bounds, which are stated here using $\mathcal{O}_p$ notation for simplicity. The statment $X = \mathcal{O}_p(A_n)$ essentially means that $X$ is $\mathcal{O}(A_n)$ with probability at least $1 - \rho$ for any constant choice of failure probability $\rho > 0$.

The full version in the supplementary material is also ambivalent to the choice of covariance for the noise variable $\mathbf{n}$, allowing in particular analysis of DP-MEPF based either on one or two moments of PFs. (The full version gives a slightly more refined treatment of the two-moment case, but the difference is typically not asymptotically relevant.)

To begin, we use standard results on Gaussians to establish that the privatized MMD is close to the non-private MMD:

**Proposition 4.1.** *Given datasets $\mathcal{D}$ and $\tilde{\mathcal{D}}$, the absolute difference between the privatized and non-private squared MMDs, a random function of only $\mathbf{n}$, satisfies*

$$\left|\widetilde{\mathrm{MMD}}^2_{k_\Phi}(\mathcal{D}, \tilde{\mathcal{D}}) - \mathrm{MMD}^2_{k_\Phi}(\mathcal{D}, \tilde{\mathcal{D}})\right| = \mathcal{O}_p\left(\frac{\sigma^2}{m^2}D + \frac{\sigma}{m}\mathrm{MMD}_{k_\Phi}(\mathcal{D}, \tilde{\mathcal{D}})\right).$$

One key quantity in the bound is $\sigma/m$, the ratio of the noise scale $\sigma$ (inversely proportional to $\varepsilon$) to the number of observed (private) data points $m$. Note that $\sigma$ depends only on the given privacy level, not on $m$, so the error becomes zero as long as $m \to \infty$. In the second term, $\sigma/m$ is multiplied by the (non-private, non-squared) MMD, which is bounded for our features, but for good generators (where our optimization hopefully spends most of its time) this term will also be nearly zero. The other term accounts for adding independent noise to each of the $D$ feature dimensions; although $D$ is typically large, so is $m^2$. Having $m = 50\text{K}$ private samples, e.g. for CIFAR-10, allows for a strong error bound as long as $D\sigma^2 \ll 625\text{M}$.

The above result is for a fixed pair of datasets. Because we only add noise $\mathbf{n}$ once, across all possible comparisons, we can use this to obtain a bound uniform over all possible generator distributions, in particular implying that the minimizer of the privatized MMD approximately minimizes the original, non-private MMD:

**Proposition 4.2.** *Fix a target dataset $\mathcal{D}$. For each $\boldsymbol{\theta}$ in some set $\Theta$, fix a corresponding $\tilde{\mathcal{D}}_{\boldsymbol{\theta}}$; in particular, $\Theta = \mathbb{R}^p$ could be the set of all generator parameters, and $\tilde{\mathcal{D}}_{\boldsymbol{\theta}}$ either the outcome of running a generator $g_{\boldsymbol{\theta}}$ on a fixed set of "seeds," $\tilde{\mathcal{D}}_{\boldsymbol{\theta}} = \{g_{\boldsymbol{\theta}}(\mathbf{z}_i)\}_{i=1}^{n}$, or the full output distribution of the generator $Q_{g_{\boldsymbol{\theta}}}$. Let $\widetilde{\boldsymbol{\theta}} \in \arg\min_{\theta \in \Theta} \widetilde{\mathrm{MMD}}^2_{k_\Phi}(\mathcal{D}, \tilde{\mathcal{D}}_\theta)$ be the private minimizer, and $\widehat{\boldsymbol{\theta}} \in \arg\min_{\theta \in \Theta} \mathrm{MMD}^2_{k_\Phi}(\mathcal{D}, \tilde{\mathcal{D}}_\theta)$ the non-private minimizer. Then $\mathrm{MMD}^2_{k_\Phi}(\mathcal{D}, \tilde{\mathcal{D}}_{\widetilde{\boldsymbol{\theta}}}) - \mathrm{MMD}^2_{k_\Phi}(\mathcal{D}, \tilde{\mathcal{D}}_{\widehat{\boldsymbol{\theta}}}) = \mathcal{O}_p\left(\frac{\sigma^2 D}{m^2} + \frac{\sigma\sqrt{D}}{m}\right)$.*

The second term of this bound will generally dominate; it arises from uniformly bounding the $\frac{\sigma}{m}\mathrm{MMD}_{k_\Phi}(\mathcal{D}, \tilde{\mathcal{D}}_\theta)$ term of Proposition 4.1 over all possible $\tilde{\mathcal{D}}_\theta$. This approach, although the default way to prove this type of bound, misses that $\mathrm{MMD}_{k_\Phi}(\mathcal{D}, \tilde{\mathcal{D}}_\theta)$ is hopefully small for $\widetilde{\boldsymbol{\theta}}$ and $\widehat{\boldsymbol{\theta}}$. We can in fact take advantage of this to provide an "optimistic" rate (Srebro et al., 2010; Zhou et al., 2021) that achieves faster convergence if the generator is capable of matching the target features (an "interpolating" regime):

**Proposition 4.3.** *In the setting of Proposition 4.2,*

$$\mathrm{MMD}^2_{k_\Phi}(\mathcal{D}, \tilde{\mathcal{D}}_{\widetilde{\boldsymbol{\theta}}}) - \mathrm{MMD}^2_{k_\Phi}(\mathcal{D}, \tilde{\mathcal{D}}_{\widehat{\boldsymbol{\theta}}}) = \mathcal{O}_p\left(\frac{\sigma^2 D}{m^2} + \frac{\sigma\sqrt{D}}{m}\mathrm{MMD}_{k_\Phi}(\mathcal{D}, \tilde{\mathcal{D}}_{\widehat{\boldsymbol{\theta}}})\right).$$

Note that this bound implies the previous one, since $\mathrm{MMD}_{k_\Phi}(\mathcal{D}, \tilde{\mathcal{D}})$ is bounded. But in the case where the generator is capable of exactly matching the features of the target distribution, the second term becomes zero, and the rate with respect to $m$ is greatly improved.

In either regime, our approximate minimization of the empirical MMD is far faster than the rate at which minimizing the empirical $\mathrm{MMD}(\mathcal{D}, Q_{g_\theta})$ converges to minimizing the true, distribution-level $\mathrm{MMD}(P, Q_{g_\theta})$: the known results there (e.g. Dziugaite et al., 2015, Theorem 1) give a $1/\sqrt{m}$ rate, compared to our $1/m$ or even $1/m^2$.

We show that minimizing DP-MEPF's loss actually pays *no* asymptotic penalty for privacy (especially when a perfect generator exists), with the privacy loss dwarfed by the statistical error for large datasets; this essentially agrees with experiments (see Section 6). This is not the case for all DP methods, and other DP generation papers didn't prove any such guarantees: DP-Sinkhorn only proved privacy, and DP-MERF showed only a much weaker guarantee (its gradient is asymptotically unbiased).

## 5 RELATED WORK

Initial work on differentially private data generation assumed strong constraints on the type of data and the intended use of the released data (Snoke & Slavković, 2018; Mohammed et al., 2011; Xiao et al., 2010; Hardt et al., 2012; Zhu et al., 2017). While these studies provide theoretical guarantees on the utility of the synthetic data, they typically do not scale to our goal of large-scale image data generation.

Recently, several papers focused on discrete data generation with limited domain size (Zhang et al., 2017; Qardaji et al., 2014; Chen et al., 2015; Zhang et al., 2021). These methods learn the correlation structure of small subsets of features and privatize them in order to produce differentially private synthetic data samples. These methods often require discretization of the data and have limited scalability, so are also unsuitable for high-dimensional image data generation.

More recently, however, a new line of work has emerged that adopt the core ideas from the recent advances in deep generative models for a broad applicability of synthetic data with differential privacy constraints. The majority of this work (Xie et al., 2018; Torkzadehmahani et al., 2019; Frigerio et al., 2019; Yoon et al., 2019; Chen et al., 2020) uses generative adversarial networks (GANs; Goodfellow et al., 2014) along with some form of DP-SGD (Abadi et al., 2016). Other works in this line include PATE-GAN based on the private aggregation of teacher ensembles (Papernot et al., 2017) and variational autoencoders (Acs et al., 2018).

The closest prior work to the proposed method is DP-MERF (Harder et al., 2021), where the kernel mean embeddings are constructed using random Fourier features (Rahimi & Recht, 2008). A recent variant of DP-MERF uses Hermite

polynomial-based mean embeddings (Vinaroz et al., 2022). Unlike these methods, we use the perceptual features from a pre-trained network to construct kernel mean embeddings. Neither previous method applies to the perceptual kernels used here, so their empirical results are far worse (as we'll see shortly). Our theoretical analysis is also much more extensive: they only proved a bound on the expected error between the private and non-private empirical MMD for a fixed pair of datasets.

More recently, a similar work to DP-MERF utilizes the Sinkhorn divergence for private data generation (Cao et al., 2021), which performs similarly to DP-MERF when the cost function is the L2 distance with a large regularizer. Another related work proposes to use the characteristic function and an adversarial re-weighting objective (Liew et al., 2022) in order to improve the generalization capability of DP-MERF.

A majority of these related methods were evaluated only on relatively simple datasets such as MNIST and FashionMNIST. Even so, the DP-GAN-based methods mostly require a large privacy budget of $\epsilon \approx 10$ to generate synthetic data samples that are reasonably close to the real data samples. Our method goes far beyond this quality with much more stringent privacy constraints, as we will now see.

## 6 EXPERIMENTS

We will now compare our method to state-of-the-art methods for DP data generation.

Table 1: Downstream accuracies by Logistic regression and MLP, evaluated on the generated data samples using MNIST and FashionMNIST as private data and SVHN and CIFAR-10 as public data, respectively. In all cases, we set $\epsilon = 10, \delta = 10^{-5}$. In our method, we used both features $\phi_1, \phi_2$.

|  |  | **DP-MEPF** | DP-Sinkhorn (Cao et al., 2021) | GS-WGAN (Chen et al., 2020) | DP-MERF (Harder et al., 2021) | DP-HP (Vinaroz et al., 2022) |
|---|---|---|---|---|---|---|
| MNIST | LogReg | **83** | 83 | 79 | 79 | 81 |
|  | MLP | **90** | 83 | 79 | 78 | 82 |
| F-MNIST | LogReg | **76** | 75 | 68 | 76 | 73 |
|  | MLP | **76** | 75 | 65 | 75 | 71 |

*Datasets.* We considered four image datasets[3] of varying complexity. We started with the commonly used datasets MNIST (LeCun & Cortes, 2010) and FashionMNIST (Xiao et al., 2017), where each consist of 60,000 $28 \times 28$ pixel grayscale images depicting hand-written digits and items of clothing, respectively, sorted into 10 classes. We also looked at the more complex CelebA (Liu et al., 2015) dataset, containing 202,599 color images of faces which we scale to sizes of $32 \times 32$ or $64 \times 64$ pixels and treat as unlabeled. We also study CIFAR-10 (Krizhevsky, 2009), a 50,000-sample dataset containing $32 \times 32$ color images of 10 classes of objects, including vehicles like ships and trucks, and animals such as horses and birds.

*Implementation.* We implemented our code for all the experiments in PyTorch (Paszke et al., 2019), using the `auto-dp` package[4] (Wang et al., 2019) for the privacy analysis. Following Harder et al. (2021), we used the generator that consists of two fully connected layers followed by two convolutional layers with bilinear upsampling, for generating both MNIST and FashionMNIST datasets. For MNIST, we used the SVHN dataset as public data to pre-train ResNet18 (He et al., 2016), from which we took the perceptual features. For FashionMNIST, we used perceptual features from a ResNet18 trained on CIFAR-10. For CelebA and CIFAR-10, we followed dos Santos et al. (2019) in using perceptual features from a pre-trained VGG (Simonyan & Zisserman, 2014) on ImageNet, and a ResNet18-based generator. Further implementation details are given in the supplementary material, which also studies how different public datasets and feature extractors impact the performance.

*Evaluation metric.* Evaluating the quality of generated data is a challenging problem of its own. We use two conventional measures. The first is the *Frechet Inception Distance (FID)* score (Heusel et al., 2017), which directly measures the quality of the generated samples. The FID score correlates with human evaluations of visual similarity to the real

---

[3]Dataset licenses: MNIST: CC BY-SA 3.0; FashionMNIST:MIT; CelebA: see https://mmlab.ie.cuhk.edu.hk/projects/CelebA.html; Cifar10: MIT

[4]https://github.com/yuxiangw/autodp

Table 2: Downstream accuracies of our method for MNIST and FashionMNIST at varying values of $\epsilon$.

| | | **MNIST** | | | | **FashionMNIST** | | | |
|---|---|---|---|---|---|---|---|---|---|
| | | $\epsilon = 5$ | $\epsilon = 2$ | $\epsilon = 1$ | $\epsilon = 0.2$ | $\epsilon = 5$ | $\epsilon = 2$ | $\epsilon = 1$ | $\epsilon = 0.2$ |
| MLP | DP-MEPF $(\phi_1, \phi_2)$ | 90 | 89 | 89 | 80 | 76 | 75 | 75 | 70 |
| | DP-MEPF $(\phi_1)$ | 88 | 88 | 87 | 77 | 75 | 76 | 75 | 69 |
| LogReg | DP-MEPF $(\phi_1, \phi_2)$ | 83 | 83 | 82 | 76 | 75 | 76 | 75 | 73 |
| | DP-MEPF $(\phi_1)$ | 81 | 80 | 79 | 72 | 75 | 76 | 76 | 72 |

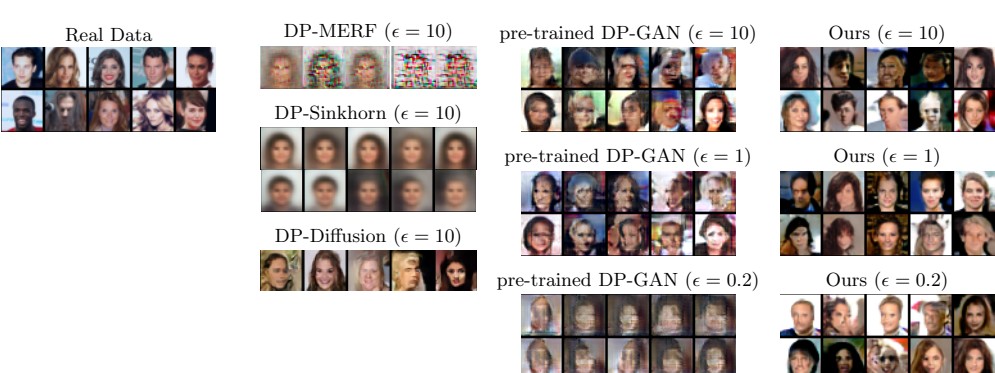

Figure 2: Synthetic $32 \times 32$ CelebA samples generated at different levels of privacy. Samples for DP-MERF and DP-Sinkhorn are taken from Cao et al. (2021) and DP-Diffusion samples are taken from Dockhorn et al. (2022). The pre-trained GAN is our baseline utilizing public data. Even at $\epsilon = 0.2$, DP-MEPF $(\phi_1, \phi_2)$ yields samples of higher visual quality than the comparison methods.

data, and is commonly used in deep generative modelling. We computed FID scores with the `pytorch_fid` package (Seitzer, 2020), based on $5\,000$ generated samples, matching dos Santos et al. (2019). As discussed in Section 3.2, we use a private proxy for FID for early stopping, while the FID scores we report in this section are non-DP measures of our final model for fair comparison to other existing methods. The second metric we use is the accuracy of downstream classifiers, trained on generated datasets and then test on the real data test sets (used by Chen et al., 2020; Torkzadehmahani et al., 2019; Yoon et al., 2019; Chen et al., 2020; Harder et al., 2021; Cao et al., 2021). This test accuracy indicates how well the downstream classifiers generalize from the synthetic to the real data distribution and thus, the utility of using synthetic data samples instead of the real ones. We computed the downstream accuracy on MNIST and FashionMNIST using the logistic regression and MLP classifiers from scikit-learn (Pedregosa et al., 2011). For CIFAR-10, we used ResNet9 taken from FFCV[5] (Leclerc et al., 2022).

In all experiments, we tested non-private training and settings with various levels of privacy, ranging from $\epsilon = 10$ (no meaningful guarantee) to $\epsilon = 0.2$ (strong privacy guarantee). We set $\delta = 10^{-5}$ for MNIST, FashionMNIST, and Cifar10 and $\delta = 10^{-6}$ for CelebA. In DP-MEPF, we also tested cases based on embeddings with only the first moment, written $(\phi_1)$, and using the first two moments, written $(\phi_1, \phi_2)$. Each value in all tables is an average of 3 or more runs; standard deviations are in the supplementary material.

Since we are unaware of any prior work on DP data generation for image data using auxiliary datasets, we instead mostly compare to recent methods which do not access auxiliary data. As expected, due to the advantage of non-private data our approach outperforms these methods by a significant margin on the more complex datasets. As a simple baseline based on public data, we also pretrain a GAN on a downscaled version of ImageNet, at $32 \times 32$, and fine-tune this model with DP-SGD on CelebA and Cifar10. We use architectures based on ResNet9 with group normalization (Wu & He, 2018) for both generator and discriminator. As suggested by Bie et al. (2023), we update the generator at a lower frequency than the discriminator and use increased minibatch sizes. Further details can be found in the supplementary material.

**MNIST and FashionMNIST.** We compare DP-MEPF to existing methods on the most common settings used in the literature, MNIST and FashionMNIST at $\epsilon = 10$, in Table 1. For an MLP on MNIST, DP-MEPF's samples far

---

[5]https://github.com/libffcv/ffcv/blob/main/examples/cifar/train_cifar.py

DP-MEPF DP-GAN

$\epsilon = 5$

$\epsilon = 1$

$\epsilon = 0.2$

Figure 3: Synthetic $64 \times 64$ CelebA samples generated at different levels of privacy with DP-MEPF $(\phi_1, \phi_2)$.

Table 3: CelebA FID scores (lower is better) for images of resolution $32 \times 32$ and $64 \times 64$. Results for DP Diffusion (DPDM) and DP Sinkhorn taken from Dockhorn et al. (2022) and Cao et al. (2021).

|    |                                | $\epsilon = 10$ | $\epsilon = 5$ | $\epsilon = 2$ | $\epsilon = 1$ | $\epsilon = 0.5$ | $\epsilon = 0.2$ |
|----|--------------------------------|------|------|------|------|-------|-------|
|    | DP-MEPF $(\phi_1, \phi_2)$     | 17.4 | 17.5 | 18.1 | 19.0 | 21.4  | 25.8  |
|    | DP-MEPF $(\phi_1)$             | 16.3 | 16.9 | 16.5 | 17.2 | 21.8  | 25.5  |
| 32 | DP-GAN (pre-trained)           | 58.1 | 66.9 | 67.1 | 81.3 | 109.1 | 192.0 |
|    | DPDM (no public data)          | 21.2 | -    | -    | 71.8 | -     | -     |
|    | DP Sinkhorn (no public data)   | 189.5| -    | -    | -    | -     | -     |
|    | DP-MEPF $(\phi_1, \phi_2)$     | 18.5 | 19.1 | 18.4 | 19.0 | 21.4  | 26.8  |
| 64 | DP-MEPF $(\phi_1)$             | 17.4 | 16.5 | 16.9 | 18.4 | 20.4  | 27.7  |
|    | DP-GAN (pre-trained)           | 57.1 | 62.3 | 65.2 | 72.5 | 91.9  | 133.3 |

outperform other methods for logistic regression and both classifiers on FashionMNIST, scores match or slightly exceed those of existing models. This might be because the domain shift between public dataset (CIFAR-10, color images of scenes) and private dataset (FashionMNIST, grayscale images of fashion items) is too large, or because the task is simple enough that random features as found in DP-MERF or DP-HP are already good enough. This will change as we proceed to more complex datasets. Table 2 shows that downstream test accuracy only starts to drop in high privacy regimes, $\epsilon < 1$, due to the low sensitivity of $\boldsymbol{\mu}_\phi$. Samples for visual comparison between methods are included in the supplementary material.

**CelebA**  Figure 2 shows that previous attempts to generate CelebA samples without auxiliary data using DP-MERF or DP-Sinkhorn have only managed to capture very basic features of the data. Each sample depicts a face, but offers no details or variety. DP-MEPF produces more accurate samples at the same $32 \times 32$ resolution, which is also reflected in improved FID scores of around 17, while DP-Sinkhorn, as reported in Cao et al. (2021), achieves an FID of 189.5. Table 3 gives FID scores for both resolutions at varying $\epsilon$. DP-MEPF consistently outperforms our pre-trained DP-GAN baseline and the scores reported for DP diffusion Dockhorn et al. (2022), As the dataset has over $200\,000$ samples, the feature embeddings have low sensitivity, and offer similar quality between $\epsilon = 10$ and $\epsilon = 1$, although quality begins to decline at $\epsilon < 1$. Samples for $64 \times 64$ images are shown in Figure 3, with similar

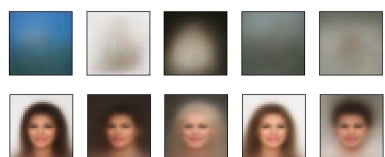

Figure 4: Samples from non-DP Sinkhorn. Top: ImageNet32. Bottom: CelebA after pretraining.

quality, and a quicker loss of quality in high privacy settings due to its larger embedding. In all cases, the $\phi_1$ embedding yields better results than $\phi_1, \phi_2$, suggesting that the second moment does not contribute useful information, perhaps because on the limited variance of the dataset.

Because DP-Sinkhorn is the best-performing method without public data, we perform experiments on DP-Sinkhorn, pretraining it non-DP on ImageNet32 and fine-tuning with DP on CelebA ($\epsilon = 10$). After seeing no improvement, we tested non-DP fine-tuning and still saw no improvements beyond what is shown in Figure 4; we tried both BigGan- and ResNet18-based generators with hyperparameter grid searches. DP-Sinkhorn only compares features at image-level, without domain-specific priors, and it appears that even non-DP the method is not powerful enough to model image data beyond MNIST. (A DP-MEPF analogue that extracts features learned from public data might help, but this would be a novel method beyond scope for comparison.) DP-MERF is similarly limited by its random features, not DP noise, as shown by non-DP versions matching $\epsilon = 10$ performance.

**Differentially private early stopping.** For CelebA and Cifar10, we use DP early stopping as explained in Section 3.2 with a privacy parameter ten times larger than the $\sigma$ used for the training objective. Keeping $(\epsilon, \delta)$ fixed, this additional

Table 4: Two examples of beneficial early stopping: For CelebA at $64 \times 64$ resolution and labeled Cifar10, DP-MEPF ($\phi_1$) sample quality (measured in FID) degrades with long training in high privacy settings (here $\epsilon \leq 1$). This makes the final model at the end of training a poor choice. Our DP selection of the best iteration via proxy stays close to the optimal choice.

|  |  | $\epsilon = 1$ | $\epsilon = 0.5$ | $\epsilon = 0.2$ |
|---|---|---|---|---|
| CelebA $64 \times 64$ | Best FID (not DP) | 17.7 | 20.1 | 27.0 |
|  | DP proxy for FID | 18.4 | 20.4 | 27.7 |
|  | At the end of training | 18.4 | 22.1 | 45.2 |
| Cifar10 (labeled) | Best FID (not DP) | 54.8 | 92.0 | 268.3 |
|  | DP proxy for FID | 56.5 | 92.0 | 268.3 |
|  | At the end of training | 198.6 | 267.7 | 357.1 |

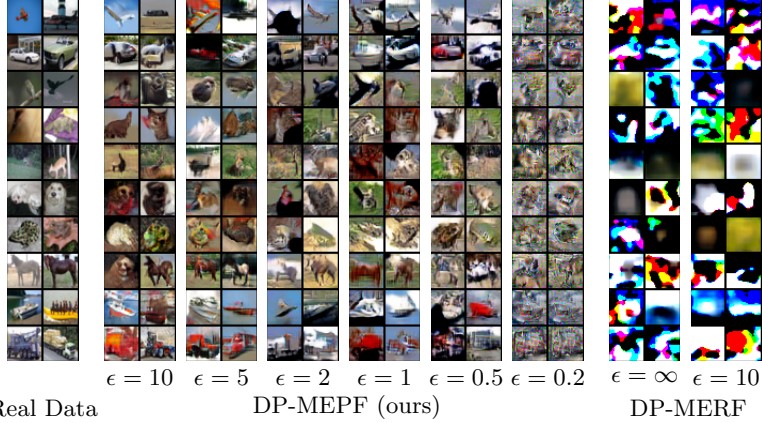

| | $\epsilon=10$ $\epsilon=5$ $\epsilon=2$ $\epsilon=1$ $\epsilon=0.5$ $\epsilon=0.2$ | $\epsilon=\infty$ $\epsilon=10$ |
| Real Data | DP-MEPF (ours) | DP-MERF |

Figure 5: Labeled samples from DP-MEPF ($\phi_1, \phi_2$) and DP-MERF (Harder et al., 2021).

release results only in a small increase in $\sigma$, and gives us a simple way for choosing the best iteration. In Table 4, we compare the true best FID, the FID picked by our private proxy, and the FID at the end of training to illustrate the advantage in high DP settings. FID scores were computed every $5\,000$ iterations, while the model trained for $200\,000$ iterations in total.

**CIFAR-10**  Finally, we investigate a dataset which has not been covered in DP data generation. While CelebA depicts a centered face in every image, CIFAR-10 includes 10 visually distinct object classes, which raises the required minimum quality of samples to somewhat resemble the dataset. At only $5\,000$ samples per class, the dataset is also significantly smaller, which poses a challenge in the private setting.

Figure 5 shows that DP-MEPF is capable of producing labelled private data (generating both labels and input images together) resembling the real data, but the quality does suffer in high privacy settings. This is also reflected in the FID scores (Table 5): at $\epsilon \leq 1$ labeled DP-MEPF scores deteriorate at a much quicker rate than the unlabeled counterpart. As the unlabeled embedding dimension is smaller by a factor of 10 (the number of classes), it is easier to release privately and retains some semblance of the data even in the highest privacy settings, as shown in Figure 6. The FID scores of our pre-trained DP-GAN baseline consistently exceed our results, usually by over 10 points. These scores are better than the DP-GAN results for CelebA, likely because $32 \times 32$ ImageNet is very similar to Cifar10. Nonetheless, the high privacy cost of DP-SGD makes DP-GAN a poor fit for a dataset of this complexity and limited size.

Table 5: FID scores for synthetic CIFAR-10 data; labeled generates both labels and images.

|  |  | $\epsilon = 10$ | $\epsilon = 5$ | $\epsilon = 2$ | $\epsilon = 1$ | $\epsilon = 0.5$ | $\epsilon = 0.2$ |
|---|---|---|---|---|---|---|---|
| unlabeled | DP-MEPF ($\phi_1, \phi_2$) | 38.8 | 37.0 | 38.7 | 43.0 | 49.4 | 67.3 |
|  | DP-MEPF ($\phi_1$) | 38.5 | 38.6 | 40.1 | 45.1 | 49.8 | 72.3 |
|  | DP-GAN | 54.6 | 54.7 | 62.4 | 74.9 | 62.7 | 73.4 |
| labeled | DP-MEPF ($\phi_1, \phi_2$) | 29.1 | 30.0 | 39.5 | 54.0 | 76.4 | 226.0 |
|  | DP-MEPF ($\phi_1$) | 30.3 | 35.6 | 42.0 | 56.5 | 92.0 | 268.3 |

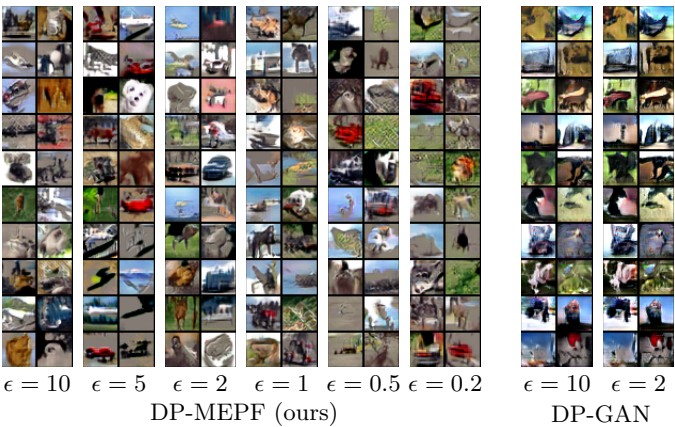

$\epsilon = 10$    $\epsilon = 5$    $\epsilon = 2$    $\epsilon = 1$    $\epsilon = 0.5$   $\epsilon = 0.2$      $\epsilon = 10$    $\epsilon = 2$

DP-MEPF (ours)                       DP-GAN

Figure 6: Unlabeled CIFAR-10 samples from DP-MEPF ($\phi_1, \phi_2$) and DP-GAN.

In Table 6 we show the test accuracy of models trained synthetic datasets applied to real data. While there is still a large gap between the 88.3% accuracy on the real data and our results, DP-MEPF achieves nontrivial results around 50% for $\epsilon = 10$, which degrade as privacy is increased. While the drop in sample quality due to high privacy is quite substantial, it is less of a problem in the unlabelled case, since our embedding dimension is smaller by a factor of 10 (the number of classes) and thus easier to release privately.

Table 6: Test accuracies (higher is better) of ResNet9 trained on CIFAR-10 synthetic data with varying privacy guarantees. When trained on real data, test accuracy is 88.3%

|  | $\epsilon = 10$ | $\epsilon = 5$ | $\epsilon = 2$ | $\epsilon = 1$ | $\epsilon = 0.5$ | $\epsilon = 0.2$ |
|---|---|---|---|---|---|---|
| **DP-MEPF** ($\phi_1, \phi_2$) | 53.0 | 43.9 | 40.0 | 28.5 | 18.0 | 16.2 |
| **DP-MEPF** ($\phi_1$) | 40.7 | 32.3 | 42.6 | 33.2 | 18.8 | 15.3 |
| DP-MERF | 13.2 | 13.4 | 13.5 | 13.8 | 13.1 | 10.4 |

# 7   DISCUSSION

We have demonstrated the advantage of using auxiliary public data in DP data generation. Our method DP-MEPF takes advantage of features from pre-trained classifiers that are readily available, and allows us to tackle datasets like CelebA and CIFAR-10, which have been unreachable for private data generation up to this point.

There are several avenues to extend our method in future work, in particular finding better options for the encoder features: the choice of VGG19 by dos Santos et al. (2019) works well in private settings, but a lower-dimensional embedding that still works well for training generative models – perhaps based on some kind of pruning scheme – might help reduce the sensitivity of $\boldsymbol{\mu}_\phi$ and improve quality.

Training other generative models such as GANs or VAEs with pretrained components is also exploring further than our initial attempt here. It may also be possible to take a "middle ground" and introduce some adaptation for features in DP-MEPF, to allow for more powerful, GAN-like models, without suffering too much privacy loss. In the non-private generative modelling community, this has proved important, but the challenge will be to do so while limiting the number of DP releases to allow modelling with, e.g., $\epsilon \leq 2$.

# 8   ACKNOWLEDGEMENTS

We thank the anonymous reviewers for their valuable time helping us improve our manuscript.

F. Harder is supported by the Max Planck Society and the Gibs Schüle Foundation and the Institutional Strategy of the University of Tübingen (ZUK63) and the German Federal Ministry of Education and Research (BMBF): Tübingen AI

Center, FKZ: 01IS18039B. F. Harder is also grateful for the support of the International Max Planck Research School for Intelligent Systems (IMPRS-IS).

M. Jalali Asadabadi, M. Park, and D. J. Sutherland are supported in part by the Natural Sciences and Engineering Research Council of Canada (NSERC) and the Canada CIFAR AI Chairs program.

## 9 BROADER IMPACT STATEMENT

Our work is motivated by the need for strong and scalable data privacy, which we expect will have mainly beneficial societal impact. However, our work touches on two topics, which are known to contain a risk of harmful impact on individuals and thus need to be treated with caution.

### 9.1 Differential privacy and fairness

Firstly, recent research has shown that DP is at odds with notions of fairness when if comes to under-represented groups in the data. For instance Chang & Shokri (2021) show that minorities are more susceptible to membership inference attacks in fair non-DP models (i.e. fairness reduces privacy) and Bagdasaryan et al. (2019) show the reverse effect: when training an unfair model with strong DP guarantees, the fairness is reduced further. The dilemma is intuitive: Fairness requires amplifying the impact of samples from minorities in the data, so they will not be ignored, while DP needs to limit the impact each individual sample can have in order to keep sensitivity low. Since its discovery, this trade-off has received attention both in works seeking a more detailed understanding (Cummings et al., 2019; Mangold et al., 2022; Esipova et al., 2022; Zhong et al., 2022; Sanyal et al., 2022) and works proposing custom approaches to DP fair machine learning (Ding et al., 2020; Xu et al., 2019; Jagielski et al., 2019; Tran et al., 2021a;b; Esipova et al., 2022). Given that the impact of DP on fairness is an active area of research and independent of our particular approach, we do not see the need to perform our own experiments on this matter.

We will, however, provide an intuition on how the problem manifests in DP-MEPF by looking at labelled data generation with significant class imbalance. Assuming an imbalanced dataset with two classes and $|C_1| = 100$ and $|C_2| = 10$, we obtain the following mean embedding:

$$\tilde{\boldsymbol{\mu}}_{\phi_t}(\mathcal{D}) = \begin{bmatrix} \frac{1}{m} \sum_{i \in C_1} \boldsymbol{\phi}_t(\mathbf{x}_i) + \mathbf{n}_{t,1} \\ \frac{1}{m} \sum_{i \in C_2} \boldsymbol{\phi}_t(\mathbf{x}_i) + \mathbf{n}_{t,2} \end{bmatrix}. \tag{7}$$

With $\|\boldsymbol{\phi}_t(\mathbf{x}_i)\|_2 = 1$, we know that the norm of the unperturbed mean embedding for class 1, given by $\|\frac{1}{m} \sum_{i \in C_1} \boldsymbol{\phi}_t(\mathbf{x}_i)\|_2 \leq 100/110$, may be ten times as large as the maximum possible norm for the class 2 embedding $\|\frac{1}{m} \sum_{i \in C_2} \boldsymbol{\phi}_t(\mathbf{x}_i)\|_2 \leq 10/110$. Nonetheless, in order to preserve DP, both embeddings are perturbed with noise of the same magnitude, leading to a significantly worse signal-to-noise ratio for the class 2 embedding. As a result, the generative model trained on this embedding will produce more accurate samples for class 1 than for class 2.

### 9.2 Differential privacy with public data

The second issue regards the use of public data in DP. In a recent position paper, Tramèr et al. (2022) raise several concerns about the increasing trend of using auxiliary datasets in DP research. Their critique has two main arguments, the first being that publicly available data may still be sensitive and using such data may cause unintended privacy violations. Given that many large datasets are scraped from the internet with limited human oversight, this data may contain personal data that was released involuntarily or shared exclusively for a specific context. The authors suggest that responsible use of public data requires improved curation practices, including e.g. collection of explicit consent for data use, auditing for and removal of sensitive content, and providing channels for reporting privacy concerns.

The other main criticism raised by Tramèr et al. (2022) is that the datasets used to demonstrate the benefits of public data in DP, such as Cifar10 or ImageNet, are poorly chosen, because they are often from nearly the same distribution as the private data. In contrast, they argue, using public data in realistic application scenarios such as medical imaging would likely require considerable domain shift, since no public data close to the target domain is available. This disparity leads to overly optimistic claims, as the experiments don't actually demonstrate good performance under significant domain shift. They further point out that the quality of a DP method becomes difficult to measure if it builds on e.g. a

non-privately pre-trained model, as overall improvements may stem both from either the private and the non-private part of the method. The authors propose dedicated benchmarks for DP machine learning should be developed, in order to obtain results which are comparable and predictive of model performance in real-world applications. They also acknowledge that such benchmarks don't currently exist and their design requires careful consideration.

We agree with the authors in their analysis of the challenges facing DP machine learning research and value their proposals for future directions and experiment design. In the light of all these problems introduced by public data, one might ask whether this is at all a research direction worth pursuing. Here, we emphasize a fact that is acknowledged in the final paragraph of Tramèr et al. (2022): *"many recent works employing public data have played an important role in showing that differential privacy can be preserved for certain complex machine learning problems, without suffering devastating impacts on utility."* DP currently sees little to no practical application in machine learning, in large part because the loss of utility it causes is often unacceptable. Auxiliary public data is the best candidate for achieving sufficient utility for practical use and so, in our eyes, the potential of these approaches outweighs the complications they introduce. It is thus vital that research in DP ML with public data is pursued further.

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

# Supplementary Material

## A Proofs

We will conduct our analysis in terms of general noise covariance $\Sigma$ for the added noise, $\mathbf{n} \sim \mathcal{N}(0, \Sigma)$. The results will depend on various norms of $\Sigma$, as well as $\|\Sigma^{1/2}\mathbf{a}\|$, where $\mathbf{a} = \boldsymbol{\mu}_\phi(\mathcal{D}) - \boldsymbol{\mu}_\phi(\tilde{\mathcal{D}})$ is the difference between empirical mean embeddings $\boldsymbol{\mu}_\phi(\mathcal{D}) = \frac{1}{|\mathcal{D}|} \sum_{\mathbf{x} \in \mathcal{D}} \phi(\mathbf{x})$. (Recall that $\mathrm{MMD}(\mathcal{D}, \tilde{\mathcal{D}}) = \|\mathbf{a}\|$.)

When we use only normalized first-moment features, the quantities appearing in the bounds are

$$\Sigma = \frac{4\sigma^2}{m^2} I_D$$

$$\|\Sigma\|_{op} = \frac{4\sigma^2}{m^2} \qquad \|\Sigma\|_F = \frac{4\sigma^2}{m^2}\sqrt{D} \qquad \mathrm{Tr}(\Sigma) = \frac{4\sigma^2}{m^2}D \tag{8}$$

$$\|\Sigma^{1/2}\mathbf{a}\|_2 = \sqrt{\mathbf{a}^\top \Sigma \mathbf{a}} = \frac{2\sigma}{m}\,\mathrm{MMD}_{k_\phi}(\mathcal{D}, \tilde{\mathcal{D}}).$$

When we use first- and second-moment features with respective scales $C_1$ and $C_2$ (both 1 in our experiments here), we have

$$\Sigma = \begin{bmatrix} \sigma^2 \left(\frac{2C_1}{m}\right)^2 I_D & 0 \\ 0 & \sigma^2 \left(\frac{2C_2}{m}\right)^2 I_D \end{bmatrix} = \frac{4\sigma^2}{m^2} \begin{bmatrix} C_1^2 I_D & 0 \\ 0 & C_2^2 I_D \end{bmatrix}$$

$$\|\Sigma\|_{op} = \frac{4\sigma^2}{m^2}\max(C_1^2, C_2^2) \quad \|\Sigma\|_F = \frac{4\sigma^2}{m^2}(C_1^2 + C_2^2)\sqrt{D} \quad \mathrm{Tr}(\Sigma) = \frac{4\sigma^2}{m^2}(C_1^2 + C_2^2)D \tag{9}$$

$$\|\Sigma^{1/2}\mathbf{a}\|_2 = \sqrt{\mathbf{a}^\top \Sigma \mathbf{a}} = \frac{2\sigma}{m}\sqrt{C_1^2\,\mathrm{MMD}\,k_{\phi_1}(\mathcal{D}, \tilde{\mathcal{D}})^2 + C_2^2\,\mathrm{MMD}\,k_{\phi_2}(\mathcal{D}, \tilde{\mathcal{D}})^2}.$$

Note that if $C_1 = C_2 = C$, then

$$\sqrt{C_1^2\,\mathrm{MMD}\,k_{\phi_1}(\mathcal{D}, \tilde{\mathcal{D}})^2 + C_2^2\,\mathrm{MMD}\,k_{\phi_2}(\mathcal{D}, \tilde{\mathcal{D}})^2} = C\,\mathrm{MMD}_{k_\Phi}(\mathcal{D}, \tilde{\mathcal{D}}).$$

### A.1 Mean absolute error of loss function

**Proposition A.1.** *Given datasets $\mathcal{D} = \{\mathbf{x}_i\}_{i=1}^m$ and $\tilde{\mathcal{D}} = \{\tilde{\mathbf{x}}_j\}_{j=1}^n$ and a kernel $k_\phi$ with a $D$-dimensional embedding $\phi$, let $\mathbf{a} = \mu_\phi(\mathcal{D}) - \mu_\phi(\tilde{\mathcal{D}})$. Define $\widetilde{\mathrm{MMD}}^2_{k_\Phi}(\mathcal{D}, \tilde{\mathcal{D}}) = \|\mathbf{a} + \mathbf{n}\|^2$ for a noise vector $\mathbf{n} \sim \mathcal{N}(0, \Sigma)$. Introducing the noise $\mathbf{n}$ affects the expected absolute error as*

$$\mathbb{E}_{\mathbf{n}}\left[\left|\widetilde{\mathrm{MMD}}^2_{k_\Phi}(\mathcal{D}, \tilde{\mathcal{D}}) - \mathrm{MMD}^2_{k_\Phi}(\mathcal{D}, \tilde{\mathcal{D}})\right|\right] \le \mathrm{Tr}(\Sigma) + 2\sqrt{\frac{2}{\pi}}\|\Sigma^{1/2}\mathbf{a}\|. \tag{10}$$

*Proof.* We have that

$$\mathbb{E}_{\mathbf{n}}\left[\left|\widetilde{\mathrm{MMD}}^2_{k_\Phi}(\mathcal{D}, \tilde{\mathcal{D}}) - \mathrm{MMD}^2_{k_\Phi}(\mathcal{D}, \tilde{\mathcal{D}})\right|\right]$$

$$= \mathbb{E}_{\mathbf{n}}\left[\left|\,\|\mathbf{a} + \mathbf{n}\|^2 - \|\mathbf{a}\|^2\,\right|\right] = \mathbb{E}_{\mathbf{n}}\left[\left|\mathbf{n}^\top \mathbf{n} + 2\mathbf{n}^\top \mathbf{a}\right|\right] \le \mathbb{E}_{\mathbf{n}}\left[\mathbf{n}^\top \mathbf{n}\right] + 2\,\mathbb{E}_{\mathbf{n}}\left[\left|\mathbf{n}^\top \mathbf{a}\right|\right]. \tag{11}$$

The first term is standard:

$$\mathbb{E}\,\mathbf{n}^\top \mathbf{n} = \mathbb{E}\,\mathrm{Tr}(\mathbf{n}^\top \mathbf{n}) = \mathbb{E}\,\mathrm{Tr}(\mathbf{n}\mathbf{n}^\top) = \mathrm{Tr}(\mathbb{E}\,\mathbf{n}\mathbf{n}^\top) = \mathrm{Tr}(\Sigma).$$

For the second, note that

$$\mathbf{a}^\top \mathbf{n} \sim \mathcal{N}(0, \mathbf{a}^\top \Sigma \mathbf{a}),$$

and so its absolute value is $\sqrt{\mathbf{a}^\top \Sigma \mathbf{a}}$ times a $\chi(1)$ random variable. Since the mean of a $\chi(1)$ distribution is $\frac{\sqrt{2}\,\Gamma(1)}{\Gamma(1/2)} = \sqrt{\frac{2}{\pi}}$, we obtain the desired bound. $\qquad\square$

## A.2 High-probability bound on the error

**Proposition A.2.** *Given datasets $\mathcal{D} = \{\mathbf{x}_i\}_{i=1}^m$ and $\tilde{\mathcal{D}} = \{\tilde{\mathbf{x}}_j\}_{j=1}^n$, let $\mathbf{a} = \mu_\phi(\mathcal{D}) - \mu_\phi(\tilde{\mathcal{D}})$, and define $\widetilde{\mathrm{MMD}}_{k_\Phi}^2(\mathcal{D}, \tilde{\mathcal{D}}) = \|\mathbf{a} + \mathbf{n}\|^2$ for a noise vector $\mathbf{n} \sim \mathcal{N}(0, \Sigma)$. Then for any $\rho \in (0, 1)$, it holds with probability at least $1 - \rho$ over the choice of $\mathbf{n}$ that*

$$\left| \widetilde{\mathrm{MMD}}_{k_\Phi}^2(\mathcal{D}, \tilde{\mathcal{D}}) - \mathrm{MMD}_{k_\Phi}^2(\mathcal{D}, \tilde{\mathcal{D}}) \right|$$
$$\leq \mathrm{Tr}(\Sigma) + \sqrt{\tfrac{2}{\pi}}\|\Sigma^{\frac{1}{2}}\mathbf{a}\|_2 + 2\left(\|\Sigma\|_F + \sqrt{2}\|\Sigma^{\frac{1}{2}}\mathbf{a}\|_2\right)\sqrt{\log(\tfrac{2}{\rho})} + 2\|\Sigma\|_{op}\log(\tfrac{2}{\rho}). \quad (12)$$

*This implies that*

$$\left| \widetilde{\mathrm{MMD}}_{k_\Phi}^2(\mathcal{D}, \tilde{\mathcal{D}}) - \mathrm{MMD}_{k_\Phi}^2(\mathcal{D}, \tilde{\mathcal{D}}) \right| = \mathcal{O}_p\left(\mathrm{Tr}(\Sigma) + \|\Sigma^{1/2}\mathbf{a}\|_2\right).$$

*Proof.* Introduce $\mathbf{z} \sim \mathcal{N}(0, I)$ such that $\mathbf{n} = \Sigma^{\frac{1}{2}}\mathbf{z}$ into Equation 11:

$$\left| \widetilde{\mathrm{MMD}}_{k_\Phi}^2(\mathcal{D}, \tilde{\mathcal{D}}) - \mathrm{MMD}_{k_\Phi}^2(\mathcal{D}, \tilde{\mathcal{D}}) \right| \leq \mathbf{n}^\top\mathbf{n} + 2\left|\mathbf{n}^\top\mathbf{a}\right| = \mathbf{z}^\top\Sigma\mathbf{z} + 2\left|\mathbf{a}^\top\Sigma^{1/2}\mathbf{z}\right|. \quad (13)$$

For the first term, denoting the eigendecomposition of $\Sigma$ as $\mathbf{Q}\boldsymbol{\Lambda}\mathbf{Q}^\top$, we can write

$$\mathbf{z}^\top\Sigma\mathbf{z} = (\mathbf{Q}^\top\mathbf{z})^\top\boldsymbol{\Lambda}(\mathbf{Q}^\top\mathbf{z}),$$

in which $\mathbf{Q}^\top\mathbf{z} \sim \mathcal{N}(0, I)$ and $\boldsymbol{\Lambda}$ is diagonal. Thus, applying Lemma 1 of Laurent & Massart (2000), we obtain that with probability at least $1 - \tfrac{\rho}{2}$,

$$\mathbf{z}^\top\Sigma\mathbf{z} \leq \mathrm{Tr}(\Sigma) + 2\|\Sigma\|_F\sqrt{\log(\tfrac{2}{\rho})} + 2\|\Sigma\|_{op}\log(\tfrac{2}{\rho}). \quad (14)$$

In the second term, $\left|\mathbf{a}^\top\Sigma^{\frac{1}{2}}\mathbf{z}\right|$, can be viewed as a function of a standard normal variable $\mathbf{z}$ with Lipschitz constant at most $\|\Sigma^{\frac{1}{2}}\mathbf{a}\|_2$. Thus, applying the standard Gaussian Lipschitz concentration inequality (Boucheron et al., 2013, Theorem 5.6), we obtain that with probability at least $1 - \tfrac{\rho}{2}$,

$$\left|\mathbf{z}^\top\Sigma^{\frac{1}{2}}\mathbf{a}\right| \leq \mathbb{E}\left|\mathbf{z}^\top\Sigma^{\frac{1}{2}}\mathbf{a}\right| + \|\Sigma^{\frac{1}{2}}\mathbf{a}\|_2\sqrt{2\log(\tfrac{2}{\rho})} = \|\Sigma^{\frac{1}{2}}\mathbf{a}\|_2\left(\sqrt{\tfrac{2}{\pi}} + \sqrt{2\log(\tfrac{2}{\rho})}\right).$$

The first statement in the theorem follows by a union bound. The $\mathcal{O}_p$ form follows by Lemma A.1 and the fact that $\mathrm{Tr}(A) \geq \|A\|_F \geq \|A\|_{op}$ for positive semi-definite matrices $A$. $\qquad\square$

The following lemma shows how to convert high-probability bounds with both sub-exponential and sub-Gaussian tails into a $\mathcal{O}_p$ statement.

**Lemma A.1.** *If a sequence of random variables $X_n$ satisfies*

$$X_n \leq A_n + B_n\sqrt{\log\frac{b_n}{\rho}} + C_n\log\frac{c_n}{\rho} \qquad \text{with probability at least } 1 - \rho,$$

*then the sequence of variables $X_n$ is*

$$\mathcal{O}_p\left(\max\left(A_n, B_n\max(\sqrt{\log b_n}, 1), C_n\max(\log c_n, 1)\right)\right).$$

*Proof.* The definition of a sequence of random variables $X_n$ being $\mathcal{O}_p(Q_n)$, where $Q_n$ is a sequence of scalars, means that the sequence $\frac{X_n}{Q_n}$ is stochastically bounded: for each $\rho$, there is some constant $R_\rho$ such that $\Pr(X_n/Q_n \geq R_\rho) \leq \rho$.

Here, we have for all $n$ with probability at least $1 - \rho$ that

$$
\frac{X_n}{\max\left(A_n, B_n \max(\sqrt{\log b_n}, 1), C_n \max(\log c_n, 1)\right)} \leq \frac{A_n + B_n\sqrt{\log \frac{b_n}{\rho}} + C_n \log \frac{c_n}{\rho}}{\max\left(A_n, B_n \max(\sqrt{\log b_n}, 1), C_n \max(\log c_n, 1)\right)}
$$

$$
= \frac{A_n + B_n\sqrt{\log b_n + \log \frac{1}{\rho}} + C_n \left[\log c_n + \log \frac{1}{\rho}\right]}{\max\left(A_n, B_n \max(\sqrt{\log b_n}, 1), C_n \max(\log c_n, 1)\right)}
$$

$$
\leq \frac{A_n + B_n\sqrt{\log b_n} + B_n\sqrt{\log \frac{1}{\rho}} + C_n \log c_n + C_n \log \frac{1}{\rho}}{\max\left(A_n, B_n \max(\sqrt{\log b_n}, 1), C_n \max(\log c_n, 1)\right)}
$$

$$
\leq 1 + 1 + \sqrt{\log \frac{1}{\rho}} + 1 + \log \frac{1}{\rho}.
$$

Thus the desired bound holds with $R_\rho = 3 + \sqrt{\log \frac{1}{\rho}} + \log \frac{1}{\rho}$. $\qquad \square$

### A.3 Quality of the private minimizer: worst-case analysis

We first show uniform convergence of the privatized MMD to the non-private MMD.

**Proposition A.3.** *Suppose that $\Phi : \mathcal{X} \to \mathbb{R}^D$ is such that $\sup_x \|\Phi(x)\| \leq B$, and let $\widetilde{\mathrm{MMD}}_{k_\Phi}(\mathcal{D}, \tilde{\mathcal{D}}) = \|\mu_\Phi(\mathcal{D}) - \mu_\Phi(\tilde{\mathcal{D}}) + \mathbf{n}\|$ for $\mathbf{n} \sim \mathcal{N}(0, \Sigma)$. Then, with probability at least $1 - \rho$ over the choice of $\mathbf{n}$,*

$$
\sup_{\mathcal{D}, \tilde{\mathcal{D}}} \left| \widetilde{\mathrm{MMD}}_{k_\Phi}^2(\mathcal{D}, \tilde{\mathcal{D}}) - \mathrm{MMD}_{k_\Phi}^2(\mathcal{D}, \tilde{\mathcal{D}}) \right|
$$

$$
\leq \mathrm{Tr}(\Sigma) + 4B\sqrt{\mathrm{Tr}(\Sigma)} + 2\left(\|\Sigma\|_F + 2B\|\Sigma\|_{op}^{\frac{1}{2}}\right)\sqrt{\log(\tfrac{2}{\rho})} + 2\|\Sigma\|_{op}\log(\tfrac{2}{\rho}) = \mathcal{O}_p\left(\mathrm{Tr}(\Sigma) + B\sqrt{\mathrm{Tr}(\Sigma)}\right),
$$

*where the supremum is taken over all distributions, including the empirical distribution of datasets $\mathcal{D}, \tilde{\mathcal{D}}$ of any size.*

*Proof.* Introducing $\mathbf{z} \sim \mathcal{N}(0, I_D)$ such that $\mathbf{n} = \Sigma^{1/2}\mathbf{z}$, we have that

$$
\sup_{\mathcal{D}, \tilde{\mathcal{D}}} \left| \widetilde{\mathrm{MMD}}_{k_\Phi}^2(\mathcal{D}, \tilde{\mathcal{D}}) - \mathrm{MMD}_{k_\Phi}^2(\mathcal{D}, \tilde{\mathcal{D}}) \right| \leq \sup_{\mathcal{D}, \tilde{\mathcal{D}}} \mathbf{z}^\top \Sigma \mathbf{z} + 2\left|\mathbf{a}^\top \Sigma^{1/2}\mathbf{z}\right|
$$

$$
\leq \mathbf{z}^\top \Sigma \mathbf{z} + 2 \sup_{\mathbf{a}:\|\mathbf{a}\|\leq 2B} \left|\mathbf{a}^\top \Sigma^{1/2}\mathbf{z}\right|
$$

$$
\leq \mathbf{z}^\top \Sigma \mathbf{z} + 2 \sup_{\mathbf{a}:\|\mathbf{a}\|\leq 2B} \|\mathbf{a}\|\|\Sigma^{1/2}\mathbf{z}\|
$$

$$
= \mathbf{z}^\top \Sigma \mathbf{z} + 4B\|\Sigma^{1/2}\mathbf{z}\|.
$$

To apply Gaussian Lipschitz concentration, we also need to know that

$$
\mathbb{E}\|\Sigma^{1/2}\mathbf{z}\| \leq \sqrt{\mathbb{E}\|\Sigma^{1/2}\mathbf{z}\|^2} = \sqrt{\mathrm{Tr}(\Sigma)};
$$

the exact expectation of a $\chi$ variable with more than one degree of freedom is inconvenient, but the gap is generally not asymptotically significant. Then we get that, with probability at least $1 - \frac{\rho}{2}$,

$$
\|\Sigma^{1/2}\mathbf{z}\| \leq \sqrt{\mathrm{Tr}(\Sigma)} + \|\Sigma\|_{op}^{1/2}\sqrt{2\log \frac{2}{\rho}}.
$$

Again combining with the bound of Equation 14, we get the stated bound. $\qquad \square$

This bound is looser than in Proposition A.2, since the term depending on $\mathbf{a}$ is now "looking at" $\mathbf{z}$ in many directions rather than just one: we end up with a $\chi(\dim(\Sigma))$ random variable instead of $\chi(1)$.

We can use this uniform convergence bound to show that the minimizer of the private loss approximately minimizes the non-private loss:

**Proposition A.4.** *Fix a target dataset $\mathcal{D}$. For each $\boldsymbol{\theta}$ in some set $\Theta$, fix a corresponding $\tilde{\mathcal{D}}_{\boldsymbol{\theta}}$; in particular, $\Theta = \mathbb{R}^p$ could be the set of all generator parameters, and $\tilde{\mathcal{D}}_{\boldsymbol{\theta}}$ either the outcome of running a generator $g_{\boldsymbol{\theta}}$ on a fixed set of "seeds," $\tilde{\mathcal{D}}_{\boldsymbol{\theta}} = \{g_{\boldsymbol{\theta}}(\mathbf{z}_i)\}_{i=1}^n$, or the full output distribution of the generator $Q_{g_{\boldsymbol{\theta}}}$. Suppose that $\Phi : \mathcal{X} \to \mathbb{R}^D$ is such that $\sup_x \|\Phi(x)\| \leq B$, and let $\widetilde{\mathrm{MMD}}_{k_\Phi}(\mathcal{D}, \tilde{\mathcal{D}}) = \|\mu_\Phi(\mathcal{D}) - \mu_\Phi(\tilde{\mathcal{D}}) + \mathbf{n}\|$ for $\mathbf{n} \sim \mathcal{N}(0, \Sigma)$. Let $\widetilde{\boldsymbol{\theta}} \in \arg\min_{\theta \in \Theta} \widetilde{\mathrm{MMD}}_{k_\Phi}^2(\mathcal{D}, \tilde{\mathcal{D}}_{\boldsymbol{\theta}})$ be the private minimizer, and $\widehat{\boldsymbol{\theta}} \in \arg\min_{\theta \in \Theta} \widehat{\mathrm{MMD}}_{k_\Phi}^2(\mathcal{D}, \tilde{\mathcal{D}}_{\theta})$ the non-private minimizer. For any $\rho \in (0, 1)$, with probability at least $1 - \rho$ over the choice of $\mathbf{n}$,*

$$\mathrm{MMD}_{k_\Phi}^2(\mathcal{D}, \tilde{\mathcal{D}}_{\widetilde{\boldsymbol{\theta}}}) - \mathrm{MMD}_{k_\Phi}^2(\mathcal{D}, \tilde{\mathcal{D}}_{\widehat{\boldsymbol{\theta}}})$$
$$\leq 2\mathrm{Tr}(\Sigma) + 8B\sqrt{\mathrm{Tr}(\Sigma)} + 4\left(\|\Sigma\|_F + 2B\|\Sigma\|_{op}^{\frac{1}{2}}\right)\sqrt{\log(\tfrac{2}{\rho})} + 4\|\Sigma\|_{op}\log(\tfrac{2}{\rho}) = \mathcal{O}_p\left(\mathrm{Tr}(\Sigma) + B\sqrt{\mathrm{Tr}(\Sigma)}\right).$$

*Proof.* Let $\alpha$ represent the uniform error bound of Proposition A.2. Applying Proposition A.2, the definition of $\widetilde{\boldsymbol{\theta}}$, then Proposition A.2 again:

$$\mathrm{MMD}_{k_\Phi}^2(\mathcal{D}, \tilde{\mathcal{D}}_{\widetilde{\boldsymbol{\theta}}}) \leq \widetilde{\mathrm{MMD}}_{k_\Phi}^2(\mathcal{D}, \tilde{\mathcal{D}}_{\widetilde{\boldsymbol{\theta}}}) + \alpha \leq \widetilde{\mathrm{MMD}}_{k_\Phi}^2(\mathcal{D}, \tilde{\mathcal{D}}_{\widehat{\boldsymbol{\theta}}}) + \alpha \leq \mathrm{MMD}_{k_\Phi}^2(\mathcal{D}, \tilde{\mathcal{D}}_{\widehat{\boldsymbol{\theta}}}) + 2\alpha. \qquad \square$$

### A.4 Quality of the private minimizer: "optimistic" analysis

The preceding analysis is quite "worst-case," since we upper-bounded the MMD by the maximum possible value everywhere. Noticing that the approximation in Proposition A.2 is tighter when $\|\Sigma^{1/2}\mathbf{a}\|$ is smaller, we can instead show an "optimistic" rate which takes advantage of this fact to show tighter approximation for the minimizer of the noised loss. In the "interpolating" case where the generator can achieve zero empirical MMD, the convergence rate substantially improves (generally improving the squared MMD from $\mathcal{O}_p(1/m)$ to $\mathcal{O}_p(1/m^2)$).

**Proposition A.5.** *In the setup of Proposition A.4, we have with probability at least $1 - \rho$ over $\mathbf{n}$ that*

$$\mathrm{MMD}_{k_\Phi}^2(\mathcal{D}, \tilde{\mathcal{D}}_{\widetilde{\boldsymbol{\theta}}}) - \mathrm{MMD}_{k_\Phi}^2(\mathcal{D}, \tilde{\mathcal{D}}_{\widehat{\boldsymbol{\theta}}})$$
$$\leq 9\mathrm{Tr}(\Sigma) + 4\sqrt{\mathrm{Tr}(\Sigma)}\,\mathrm{MMD}_{k_\Phi}(\mathcal{D}, \tilde{\mathcal{D}}_{\widehat{\boldsymbol{\theta}}})$$
$$+ 2\left(9\|\Sigma\|_F + 2\sqrt{2\|\Sigma\|_{op}}\,\mathrm{MMD}_{k_\Phi}(\mathcal{D}, \tilde{\mathcal{D}}_{\widehat{\boldsymbol{\theta}}})\right)\sqrt{\log\frac{2}{\rho}} + 18\|\Sigma\|_{op}\log\frac{2}{\rho}$$
$$= \mathcal{O}_p\left(\mathrm{Tr}(\Sigma) + \sqrt{\mathrm{Tr}(\Sigma)}\,\mathrm{MMD}_{k_\Phi}(\mathcal{D}, \tilde{\mathcal{D}}_{\widehat{\boldsymbol{\theta}}})\right).$$

*Proof.* Let's use $\widehat{\mathrm{MMD}}(\boldsymbol{\theta})$ to denote $\mathrm{MMD}_{k_\Phi}(\mathcal{D}, \tilde{\mathcal{D}}_{\boldsymbol{\theta}})$, and $\widetilde{\mathrm{MMD}}(\boldsymbol{\theta})$ for $\widetilde{\mathrm{MMD}}_{k_\Phi}(\mathcal{D}, \tilde{\mathcal{D}}_{\boldsymbol{\theta}})$.

For all $\boldsymbol{\theta}$, we have that

$$\left|\widetilde{\mathrm{MMD}}^2(\boldsymbol{\theta}) - \widehat{\mathrm{MMD}}^2(\boldsymbol{\theta})\right| \leq \mathbf{z}^\top\Sigma\mathbf{z} + 2\left|(\mu^\Phi(\mathcal{D}) - \mu^\Phi(\tilde{\mathcal{D}}))^\top\Sigma^{1/2}\mathbf{z}\right|$$
$$\leq \mathbf{z}^\top\Sigma\mathbf{z} + 2\widehat{\mathrm{MMD}}(\boldsymbol{\theta})\|\Sigma^{1/2}\mathbf{z}\|.$$

Thus, applying this inequality in both the first and third lines,

$$\widehat{\mathrm{MMD}}^2(\widetilde{\boldsymbol{\theta}}) \leq \widetilde{\mathrm{MMD}}^2(\widetilde{\boldsymbol{\theta}}) + \mathbf{z}^\top\Sigma\mathbf{z} + 2\widehat{\mathrm{MMD}}(\widetilde{\boldsymbol{\theta}})\|\Sigma^{1/2}\mathbf{z}\|$$
$$\leq \widetilde{\mathrm{MMD}}^2(\widehat{\boldsymbol{\theta}}) + \mathbf{z}^\top\Sigma\mathbf{z} + 2\widehat{\mathrm{MMD}}(\widetilde{\boldsymbol{\theta}})\|\Sigma^{1/2}\mathbf{z}\|$$
$$\leq \widehat{\mathrm{MMD}}^2(\widehat{\boldsymbol{\theta}}) + 2\mathbf{z}^\top\Sigma\mathbf{z} + 2\left(\widehat{\mathrm{MMD}}(\widetilde{\boldsymbol{\theta}}) + \widehat{\mathrm{MMD}}(\widehat{\boldsymbol{\theta}})\right)\|\Sigma^{1/2}\mathbf{z}\|;$$

in the second line we used that $\widetilde{\mathrm{MMD}}(\widetilde{\boldsymbol{\theta}}) \leq \widetilde{\mathrm{MMD}}(\widehat{\boldsymbol{\theta}})$. Rearranging, we get that

$$\widehat{\mathrm{MMD}}^2(\widetilde{\boldsymbol{\theta}}) - \beta\,\widehat{\mathrm{MMD}}(\widetilde{\boldsymbol{\theta}}) - \gamma \leq 0, \tag{15}$$

where

$$\beta = 2\|\Sigma^{1/2}\mathbf{z}\| \geq 0$$

$$\gamma = \widehat{\mathrm{MMD}}^2(\widehat{\boldsymbol{\theta}}) + 2\mathbf{z}^\top \Sigma \mathbf{z} + 2\widehat{\mathrm{MMD}}(\widehat{\boldsymbol{\theta}})\|\Sigma^{1/2}\mathbf{z}\| \geq 0.$$

The left-hand side of Equation 15 is a quadratic in $\widehat{\mathrm{MMD}}(\tilde{\boldsymbol{\theta}})$ with positive curvature; it has two roots, at

$$\frac{\beta}{2} \pm \sqrt{\left(\frac{\beta}{2}\right)^2 + \gamma}.$$

Thus the inequality Equation 15 can only hold in between the roots; the root with a minus sign is negative, and so does not concern us since we know that $\widehat{\mathrm{MMD}}(\boldsymbol{\theta}) \geq 0$. Thus, for Equation 15 to hold, we must have

$$\widehat{\mathrm{MMD}}(\widetilde{\boldsymbol{\theta}}) \leq \tfrac{\beta}{2} + \sqrt{\left(\tfrac{\beta}{2}\right)^2 + \gamma}$$

$$\widehat{\mathrm{MMD}}^2(\widetilde{\boldsymbol{\theta}}) \leq \tfrac{\beta^2}{4} + \left(\tfrac{\beta}{2}\right)^2 + \gamma + \beta\sqrt{\left(\tfrac{\beta}{2}\right)^2 + \gamma}$$

$$\leq \gamma + \beta^2 + \beta\sqrt{\gamma}.$$

Also note that

$$\gamma = \widehat{\mathrm{MMD}}^2(\widehat{\boldsymbol{\theta}}) + 2\mathbf{z}^\top \Sigma \mathbf{z} + 2\widehat{\mathrm{MMD}}(\widehat{\boldsymbol{\theta}})\|\Sigma^{1/2}\mathbf{z}\| \leq \left(\widehat{\mathrm{MMD}}(\widehat{\boldsymbol{\theta}}) + \sqrt{2}\|\Sigma^{1/2}\mathbf{z}\|\right)^2.$$

Thus, substituting in for $\beta$ and $\gamma$ then simplifying, we have that

$$\widehat{\mathrm{MMD}}^2(\widetilde{\boldsymbol{\theta}}) \leq \widehat{\mathrm{MMD}}^2(\widehat{\boldsymbol{\theta}}) + (6 + 2\sqrt{2})\mathbf{z}^\top \Sigma \mathbf{z} + 4\|\Sigma^{1/2}\mathbf{z}\|\,\widehat{\mathrm{MMD}}(\widehat{\boldsymbol{\theta}}).$$

Using the same bounds on $\mathbf{z}^\top \Sigma \mathbf{z}$ and $\|\Sigma^{1/2}\mathbf{z}\|$ as in Proposition A.3, and $6\sqrt{2} < 9$, gives the claimed bound. $\qquad\square$

## B   Extended Implementation details

**Repository.**   Our code is available at `https://github.com/ParkLabML/DP-MEPF`; the readme files contain further instructions on how to run the code.

### B.1   Hyperparameter settings

For each dataset, we tune the generator learning rate ($\mathrm{LR}_{gen}$) and moving average learning rate ($\mathrm{LR}_{mavg}$) from choices $10^{-k}$ and $3 \cdot 10^{-k}$ with $k \in \{3, 4, 5\}$ once for the non-private setting and once at $\epsilon = 2$. The latter is used in all private experiments for that dataset, as shown in 7. After some initial unstructured experimentation, hyperparameters are chosen with identical values across dataset shown in 8

For the Cifar10 DP-MERF baseline we tested random tuned random features dimension $d \in \{10000, 50000\}$, random features sampling distribution $\sigma \in \{100, 300, 1000\}$, learning rate decay by 10% every $e \in \{1000, 10000\}$ iterations and learning rate $10^{-k}$ with $k \in \{2, 3, 4, 5, 6\}$. Results presented use $d = 500000, \sigma = 1000, e = 10000, k = 3$.

The DP-GAN baseline for Cifar10 and CelebA uses the same generator as DP-MEPF with 3 residual blocks and a total of 8 convolutional layers and is paired with a ResNet9 discriminator which uses Groupnorm instead of Batchnorm to allow for per-sample gradient computation. We pre-train the model non-privately to convergence on downsampled imagenet in order to maintain the same resolution of $32 \times 32$ and then fine-tune the model for a smaller number of epochs. In case of the CelebA $64 \times 64$ data we add another residual block to discriminator and generator to account for the doubling in resolution. The base multiplier for number of feature maps is reduced from 64 to 50 to lessen the increase in number of weights. Results are the best scores of a grid-search over the following parameters at $\epsilon = 2$, which is then used in all settings: number of epochs $\{1, 10, 30, 50\}$ generator and discriminator learning rate separately for $10^{-k}$ and $3 \cdot 10^{-k}$ with $k \in \{3, 4, 5\}$, clip-norm $\{10^{-3}, 10^{-4}, 10^{-5}, 10^{-6}\}$, batch size $\{128, 256, 512\}$ and, as advised in Bie et al. (2023), number of discriminator updates per generator $\{1, 10, 30, 50\}$. The chosen values are given in table 9.

Table 7: Learning rate hyperparameters across datasets

| Dataset | $\varepsilon$ | $(\phi_1, \phi_2)$ | | $(\phi_1)$ | |
|---|---|---|---|---|---|
| | | $\text{LR}_{gen}$ | $\text{LR}_{mavg}$ | $\text{LR}_{gen}$ | $\text{LR}_{mavg}$ |
| MNIST | $\varepsilon = \infty$ | $10^{-5}$ | $10^{-3}$ | $10^{-5}$ | $10^{-3}$ |
| | $\varepsilon < \infty$ | $10^{-5}$ | $10^{-4}$ | $10^{-5}$ | $10^{-4}$ |
| FashionMNIST | $\varepsilon = \infty$ | $10^{-5}$ | $10^{-3}$ | $10^{-5}$ | $10^{-3}$ |
| | $\varepsilon < \infty$ | $10^{-4}$ | $10^{-3}$ | $10^{-4}$ | $10^{-3}$ |
| CelebA32 | $\{\infty, 10, 5\}$ | $3 \cdot 10^{-4}$ | $10^{-4}$ | $3 \cdot 10^{-4}$ | $10^{-4}$ |
| | $\{2, 1\}$ | $3 \cdot 10^{-4}$ | $3 \cdot 10^{-4}$ | $3 \cdot 10^{-4}$ | $3 \cdot 10^{-4}$ |
| | $\{0.5, 0.2\}$ | $10^{-3}$ | $3 \cdot 10^{-4}$ | $\cdot 10^{-3}$ | $3 \cdot 10^{-4}$ |
| CelebA64 | $\{\infty, 10, 5\}$ | $3 \cdot 10^{-4}$ | $10^{-4}$ | $3 \cdot 10^{-4}$ | $3 \cdot 10^{-4}$ |
| | $\{2, 1\}$ | $3 \cdot 10^{-4}$ | $10^{-3}$ | $3 \cdot 10^{-4}$ | $3 \cdot 10^{-4}$ |
| | $\{0.5, 0.2\}$ | $10^{-3}$ | $10^{-3}$ | $10^{-3}$ | $10^{-3}$ |
| Cifar10 labeled | $\{\infty, 10, 5\}$ | $10^{-3}$ | $3 \cdot 10^{-4}$ | $10^{-3}$ | $10^{-4}$ |
| | $\{2, 1\}$ | $10^{-3}$ | $10^{-2}$ | $10^{-3}$ | $10^{-2}$ |
| | $\{0.5, 0.2\}$ | $10^{-3}$ | $10^{-2}$ | $10^{-3}$ | $10^{-2}$ |
| Cifar10 unlabeled | $\{\infty, 10, 5\}$ | $10^{-3}$ | $10^{-3}$ | $10^{-3}$ | $10^{-3}$ |
| | $\{2, 1\}$ | $10^{-3}$ | $10^{-3}$ | $10^{-3}$ | $10^{-3}$ |
| | $\{0.5, 0.2\}$ | $10^{-3}$ | $10^{-3}$ | $10^{-3}$ | $10^{-3}$ |

Table 8: Hyperparameters fixed across datasets

| Parameter | Value |
|---|---|
| $(\phi_1)$-bound | 1 |
| $(\phi_2)$-bound | 1 |
| iterations (MNIST & FashionMNIST) | 100,000 |
| batch size (MNIST and FashionMNIST) | 100 |
| iterations (Cifar10 & CelebA) | 200,000 |
| batch size (Cifar10 and CelebA) | 128 |
| seeds | 1,2,3,4,5 |

# C   Detailed Tables

Below we present the results from the main paper with added $a \pm b$ notation, where $a$ is the mean and $b$ is the standard deviation of the score distribution across three independent runs for MNIST and FashionMNIST and 5 independent runs for Cifar10 and CelebA.

Table 10: Downstream accuracies of our method for MNIST at varying values of $\epsilon$

| | | $\epsilon = \infty$ | $\epsilon = 10$ | $\epsilon = 5$ | $\epsilon = 2$ | $\epsilon = 1$ | $\epsilon = 0.2$ |
|---|---|---|---|---|---|---|---|
| MLP | DP-MEPF $(\phi_1, \phi_2)$ | $91.4 \pm 0.3$ | $89.8 \pm 0.5$ | $89.9 \pm 0.2$ | $89.3 \pm 0.3$ | $89.3 \pm 0.6$ | $79.9 \pm 1.3$ |
| | DP-MEPF $(\phi_1)$ | $88.2 \pm 0.6$ | $88.8 \pm 0.1$ | $88.4 \pm 0.5$ | $88.0 \pm 0.2$ | $87.5 \pm 0.6$ | $77.1 \pm 0.4$ |
| LogReg | DP-MEPF $(\phi_1, \phi_2)$ | $84.6 \pm 0.5$ | $83.4 \pm 0.6$ | $83.3 \pm 0.7$ | $82.9 \pm 0.7$ | $82.5 \pm 0.5$ | $75.8 \pm 1.1$ |
| | DP-MEPF $(\phi_1)$ | $81.4 \pm 0.4$ | $80.8 \pm 0.9$ | $80.8 \pm 0.8$ | $80.5 \pm 0.6$ | $79.0 \pm 0.6$ | $72.1 \pm 1.4$ |

Table 9: Hyperparameters of DP-GAN for Cifar10 and CelebA

| | Cifar10 | CelebA $32 \times 32$ | CelebA $64 \times 64$ | | | |
|---|---|---|---|---|---|---|
| | | | $\epsilon \in \{0.2, 0.5\}$ | $\epsilon = 1$ | $\epsilon = 2$ | $\epsilon \in \{5, 10\}$ |
| $\text{LR}_{gen}$ | $10^{-4}$ | $3 \cdot 10^{-4}$ | $3 \cdot 10^{-4}$ | $3 \cdot 10^{-4}$ | $3 \cdot 10^{-4}$ | $3 \cdot 10^{-4}$ |
| $\text{LR}_{dis}$ | $10^{-3}$ | $3 \cdot 10^{-4}$ | $10^{-3}$ | $3 \cdot 10^{-4}$ | $10^{-3}$ | $10^{-3}$ |
| batch size | 512 | 512 | 512 | 512 | 512 | 512 |
| epochs | 10 | 10 | 10 | 10 | 10 | 10 |
| discriminator frequency | 10 | 10 | 30 | 30 | 10 | 10 |
| clip norm | $10^{-5}$ | $10^{-4}$ | $10^{-5}$ | $10^{-5}$ | $10^{-4}$ | $10^{-5}$ |

Table 11: Downstream accuracies of our method for FashionMNIST at varying values of $\epsilon$

| | | $\epsilon = \infty$ | $\epsilon = 10$ | $\epsilon = 5$ | $\epsilon = 2$ | $\epsilon = 1$ | $\epsilon = 0.2$ |
|---|---|---|---|---|---|---|---|
| MLP | DP-MEPF $(\phi_1, \phi_2)$ | $74.4 \pm 0.3$ | $76.0 \pm 0.4$ | $75.8 \pm 0.6$ | $75.1 \pm 0.3$ | $74.7 \pm 1.1$ | $70.4 \pm 1.9$ |
| | DP-MEPF $(\phi_1)$ | $73.8 \pm 0.5$ | $75.5 \pm 0.6$ | $75.1 \pm 0.8$ | $75.8 \pm 0.7$ | $75.0 \pm 1.8$ | $69.0 \pm 1.5$ |
| LogReg | DP-MEPF $(\phi_1, \phi_2)$ | $74.3 \pm 0.1$ | $75.7 \pm 1.0$ | $75.2 \pm 0.4$ | $75.8 \pm 0.4$ | $75.4 \pm 1.1$ | $72.5 \pm 1.2$ |
| | DP-MEPF $(\phi_1)$ | $72.8 \pm 0.5$ | $75.5 \pm 0.1$ | $75.5 \pm 0.8$ | $76.4 \pm 0.8$ | $76.2 \pm 0.8$ | $71.7 \pm 0.4$ |

Table 12: CelebA FID scores $32 \times 32$ (lower is better)

| | $\epsilon = \infty$ | $\epsilon = 10$ | $\epsilon = 5$ | $\epsilon = 2$ | $\epsilon = 1$ | $\epsilon = 0.5$ | $\epsilon = 0.2$ |
|---|---|---|---|---|---|---|---|
| DP-MEPF $(\phi_1, \phi_2)$ | $18.5 \pm 0.5$ | $17.4 \pm 0.7$ | $17.5 \pm 0.6$ | $18.1 \pm 0.8$ | $19.0 \pm 0.5$ | $21.4 \pm 1.3$ | $25.8 \pm 2.1$ |
| DP-MEPF $(\phi_1)$ | $16.6 \pm 0.7$ | $16.3 \pm 0.9$ | $16.9 \pm 0.5$ | $16.5 \pm 0.8$ | $17.2 \pm 0.9$ | $21.8 \pm 1.0$ | $25.5 \pm 1.1$ |

Table 13: CelebA FID scores $64 \times 64$ (lower is better)

| | $\epsilon = \infty$ | $\epsilon = 10$ | $\epsilon = 5$ | $\epsilon = 2$ | $\epsilon = 1$ | $\epsilon = 0.5$ | $\epsilon = 0.2$ |
|---|---|---|---|---|---|---|---|
| DP-MEPF $(\phi_1, \phi_2)$ | $18.6 \pm 1.0$ | $18.5 \pm 1.2$ | $19.1 \pm 0.9$ | $18.4 \pm 1.0$ | $19.0 \pm 1.2$ | $21.4 \pm 1.3$ | $26.8 \pm 1.5$ |
| DP-MEPF $(\phi_1)$ | $16.3 \pm 0.4$ | $17.4 \pm 1.4$ | $16.5 \pm 0.8$ | $16.9 \pm 1.1$ | $18.4 \pm 0.9$ | $20.4 \pm 0.8$ | $27.7 \pm 2.1$ |

Table 14: FID scores for synthetic *labelled* CIFAR-10 data (generating both labels and input images)

| | $\epsilon = \infty$ | $\epsilon = 10$ | $\epsilon = 5$ | $\epsilon = 2$ | $\epsilon = 1$ | $\epsilon = 0.5$ | $\epsilon = 0.2$ |
|---|---|---|---|---|---|---|---|
| **DP-MEPF** $(\phi_1, \phi_2)$ | $27.7 \pm 3.1$ | $29.1 \pm 1.3$ | $30.0 \pm 0.8$ | $39.5 \pm 1.9$ | $54.0 \pm 1.3$ | $76.4 \pm 3.9$ | $226.0 \pm 5.4$ |
| **DP-MEPF** $(\phi_1)$ | $28.4 \pm 2.8$ | $30.3 \pm 2.1$ | $35.6 \pm 5.8$ | $42.0 \pm 3.0$ | $56.5 \pm 3.4$ | $92.0 \pm 3.5$ | $268.3 \pm 8.5$ |

Table 15: Test accuracies (higher better) of ResNet9 trained on CIFAR-10 synthetic data with varying privacy guarantees. When trained on real data, test accuracy is 88.3%

| | $\epsilon = \infty$ | $\epsilon = 10$ | $\epsilon = 5$ | $\epsilon = 2$ | $\epsilon = 1$ | $\epsilon = 0.5$ | $\epsilon = 0.2$ |
|---|---|---|---|---|---|---|---|
| **DP-MEPF** $(\phi_1, \phi_2)$ | $57.5 \pm 3.3$ | $53.0 \pm 2.8$ | $43.9 \pm 1.2$ | $40.0 \pm 1.9$ | $28.5 \pm 4.5$ | $18.0 \pm 1.0$ | $16.2 \pm 1.8$ |
| **DP-MEPF** $(\phi_1)$ | $43.8 \pm 3.5$ | $40.7 \pm 4.2$ | $32.3 \pm 6.2$ | $42.6 \pm 1.6$ | $33.2 \pm 2.6$ | $18.8 \pm 4.0$ | $15.3 \pm 2.5$ |

## D  Encoder architecture comparison

We are testing a large collection of classifiers of different sizes from the torchvision library including VGG, ResNet, ConvNext and EfficientNet. For each we look at unlabelled Cifar10 generation quality in the non-DP setting and at $\epsilon = 0.2$. In each architecture, we use all activations from convolutional layers with a kernel size greater than 1x1. We list the number of extracted features along with the achieved FID score in table 17, where each result is the best result obtained by tuning learning rates. As already observed in dos Santos et al. (2019), we find that VGG architectures appear to learn particularly useful features for feature matching. We hypothesized that in the private setting other architectures with fewer features might outperform the VGG model, but have found this to not be the case.

## E  Public dataset comparison

We pretrained a ResNet18 using ImageNet, CIFAR10, and SVHN as our public data, respectively. We then used the perceptual features to train a generator using CelebA dataset as our private data at a privacy budget of $\epsilon = 0.2$ and obtained the scores shown in 18. These numbers reflect our intuition that as long as the public data is sufficiently similar and contains more complex patterns than private data, e.g., transferring the knowledge learned from ImageNet as public data to generate CelebA images as private data, the learned features from public data are useful enough to generate good synthetic data. In addition, as the public data become more simplistic (from CIFAR10 to SVHN), the usefulness of such features reduces in producing good CelebA synthetic samples.

Table 16: FID scores for synthetic *unlabelled* CIFAR-10 data

| | $\epsilon = \infty$ | $\epsilon = 10$ | $\epsilon = 5$ | $\epsilon = 2$ | $\epsilon = 1$ | $\epsilon = 0.5$ | $\epsilon = 0.2$ |
|---|---|---|---|---|---|---|---|
| **DP-MEPF $(\phi_1, \phi_2)$** | $38.5 \pm 1.5$ | $38.8 \pm 2.0$ | $37.0 \pm 1.1$ | $38.7 \pm 2.2$ | $43.0 \pm 1.1$ | $49.4 \pm 1.0$ | $67.3 \pm 2.6$ |
| **DP-MEPF $(\phi_1)$** | $38.5 \pm 0.6$ | $38.5 \pm 0.4$ | $38.6 \pm 1.3$ | $40.1 \pm 1.1$ | $45.1 \pm 2.4$ | $49.8 \pm 2.5$ | $72.3 \pm 4.0$ |

Table 17: Unlabeled Cifar10 FID scores achieved with different feature extractors. VGG models yield the best results in both non-DP and high DP settings.

| Encoder model | #features | $\epsilon = \infty$ | | $\epsilon = 0.2$ | |
|---|---|---|---|---|---|
| | | $(\phi_1, \phi_2)$ | $(\phi_1)$ | $(\phi_1, \phi_2)$ | $(\phi_1)$ |
| VGG19 | 303104 | 35.0 | 37.0 | 56.2 | 85.8 |
| VGG16 | 276480 | 37.4 | 39.8 | 71.4 | 72.2 |
| VGG13 | 249856 | 38.2 | 36.7 | 78.1 | 71.2 |
| VGG11 | 151552 | 40.5 | 41.6 | 65.4 | 68.6 |
| ResNet152 | 429568 | 71.8 | 70.1 | 88.6 | 87.9 |
| ResNet101 | 300544 | 77.5 | 73.7 | 76.0 | 82.4 |
| ResNet50 | 196096 | 71.5 | 76.3 | 90.0 | 105.1 |
| ResNet34 | 72704 | 74.8 | 103.3 | 89.1 | 93.1 |
| ResNet18 | 47104 | 84.9 | 85.0 | 104.5 | 95.2 |
| ConvNext large | 161280 | 141.9 | 232.0 | 138.2 | 221.6 |
| ConvNext base | 107520 | 142.4 | 248.0 | 157.0 | 200.1 |
| ConvNext small | 80640 | 171.7 | 212.3 | 169.9 | 202.9 |
| ConvNext tiny | 52992 | 145.6 | 218.2 | 138.8 | 205.8 |
| EfficientNet L | 119168 | 200.9 | 229.0 | 243.7 | 226.6 |
| EfficientNet M | 68704 | 185.7 | 177.1 | 218.7 | 227.1 |
| EfficientNet S | 47488 | 157.5 | 160.6 | 171.5 | 186.7 |

Table 18: FID scores achieved for CelebA $32 \times 32$ using a ResNet encoder with different public training sets

| | ImageNet | Cifar10 | SVHN |
|---|---|---|---|
| FID | 47.6 | 51.2 | 65.2 |

## F   Training DP-MEPF without auxiliary data

While DP-MEPF is explicitly designed to take advantage of available public data, one might wonder how the method performs if no such data is available. The following experiment on CIFAR10 explores this scenario. We assume that a privacy budget of $\epsilon = 10$ is given. We use some part of the budget for feature extractor (i.e. the classifier) training and the rest of the budget for the generator training.

For a feature extractor, we have trained ResNet-20 classifiers with DP-SGD at three different levels of $\epsilon \in \{2, 5, 8\}$ for classifying the CIFAR10 dataset. We set the clipping norm to 0.01 and trained the classifiers for $7, 49$ and $98$ epochs, respectively. Their test accuracies are $38.4\%, 49.5\%$ and $54.0\%$ respectively. We also include scores for DP-MEPF applied to the untrained Classifier, denoted as $\epsilon = 0$.

Then, we train the generator using these four sets of features to generate CIFAR10 images, where each generator training uses the rest of the budget, i.e., $\epsilon \in \{8, 5, 2\}$ and $\epsilon = 10$ for the untrained classifier. We tune the learning rate in each of the four settings and keep other hyperparameters at default values.

Table 19: DP-MEPF results in CIFAR10 when using a DP feature extractor ($\epsilon = 0$ is an untrained extractor)

| $\epsilon$ for feature extractor training | for generator training | FID |
|---|---|---|
| 0 | 10 | 111.1 |
| 2 | 8 | 127.0 |
| 5 | 5 | 90.8 |
| 8 | 2 | 119.0 |

As expected, in Table 19 we see a considerable increase in the FID score, compared to DP-MEPF with public data. A balanced allocation of privacy budget with $\epsilon = 5$ each for classifier and generator training yields the best result at an FID score of 90.8 and performs significantly better than just using a randomly initialized feature extractor, which only achieves a score of 111.1. For comparison: with public data DP-MEPF achieves an FID score of 37.0 at $\varepsilon = 5$, highlighting the importance of such data to our method.

## G   Additional Plots

Below we show samples from our generated MNIST and FashionMNIST data in Figure 7 and Figure 8 respectively.

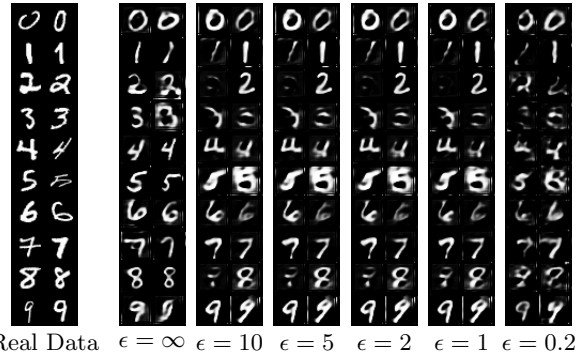

Figure 7: MNIST samples produced with DP-MEPF ($\phi_1, \phi_2$) at various levels of privacy

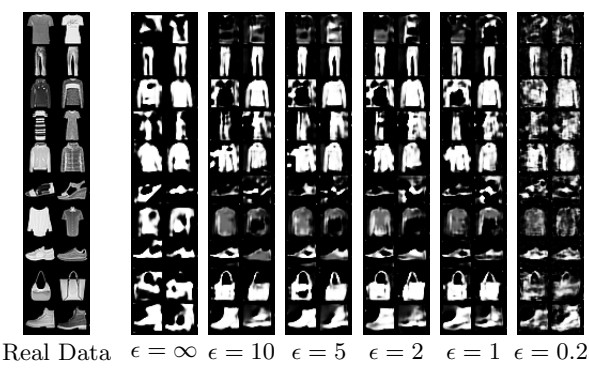

Figure 8: Fashion-MNIST samples produced with DP-MEPF ($\phi_1, \phi_2$) at various levels of privacy

