# OpenReview forum: "Pre-trained Perceptual Features Improve Differentially Private Image Generation"
_TMLR — Accepted by TMLR_

### Review · Reviewer_nrzH · 2023-02-15

**Summary Of Contributions:**

This paper proposes to generate images in a differentially private manner using a three-stage pipeline. First, the authors pre-train a feature extractor network on a public dataset. Then, using this public network, they extract the mean embedding of the private dataset (from which to generate) and privatize it using the Gaussian mechanism. Finally, using post-processing, the authors leverage this private mean embedding to train (without DP) a network that generates images with a mean embedding that is close to the privatized one (minimizing the Maximum Mean Discrepancy or MMD).

The authors claim that the proposed method allows them to generate "reasonable" CIFAR-10, CelebA, MNIST and FashionMNIST-like images with high privacy regimes ($\varepsilon = 2$ for CIFAR-10), as illustrated in Figures 2-7. They also provide a theoretical analysis showing that the noise added once to privatize the mean embedding of the private dataset does not hurt the generative model's convergence.

**Audience:**

Yes

**Claims And Evidence:**

Yes

**Requested Changes:**

In summary, the paper does match claims with evidence. We also thank the authors for providing code (which the reviewer read in part but did not execute). Note that the reviewer did not read the proofs nor check the correctness of the theoretical analysis in Section 4 but rather focused on the method, the experimental results and setup.

For a detailed list of requested changes, see the "Strengths and Weaknesses" section.

**Strengths And Weaknesses:**

Strengths:
- The proposed method abstracts away the need to inject noise directly during the generation process (for instance with a GAN or a Diffusion model) by using a privatized mean embedding of the data distribution to model.
- The authors experiment on 4 datasets and present convincing generation results for decently small values of the privacy budget epsilon.
- The proposed method (DP-MEPF) generates synthetic images that consistently degrade less the accuracy on downstream tasks than previous state-of-the-art methods.
- The authors propose a private early stopping method (Section 3.2) using a private proxy of the FID score. We thank the authors for paying attention to potential privacy leaks occurring through training hyperparameters, even though the authors could add that the bandwidth of such hyperparameters is generally too small to incur a meaningful privacy loss [1]

Weaknesses:
- The proposed method critically relies on (a tuple of) privatized mean embeddings that represent the true data distribution to model. Could the authors elaborate on the soundness of this method for modelling extremely large image datasets? Indeed, the largest considered dataset is CelebA (200k samples), whereas recent generative models leverage LAION (400 million images). In that case, would the mean embedding be enough to capture the diversity of a large-scale dataset?
- The authors could add some discussion on the assumption of a publicly available dataset. Following [2], we can question whether public datasets generally scrapped from the web could be (1) considered as privacy-preserving and (2) available for certain tasks. Moreover, acquiring information only through the DP channel is indeed more difficult but could alleviate these concerns (the reviewer does not ask for additional experiments but rather to give more substance to this discussion in the paper).
- The authors could mention recent/concurrent work on Differentially Private Stable Diffusion [3]
- The authors motivate the need for synthetic image generation mainly through the angle of downstream tasks, for instance by generating private images that still maintain the classification accuracy of a given model. However, another angle is to be able to generate images from the original distribution that are private and consider these images as the end product. In that sense, a potential follow-up would be to perform empirical attacks on the generated images to see how close to the original training set these are.

[1] "Private selection from private candidates", Liu and Talwar, 2019.

[2] "Considerations for Differentially Private Learning with Large-Scale Public Pretraining", Tramer et al, 2022

[3] "Differentially Private Diffusion Models", Dockhorn et al, 2022

---

> ### Author Response · Authors · 2023-03-10
> **Thank you for your feedback!**
>
> Below, we answer your individual questions in detail or refer to the relevant section in the updated document. If one of your concerns is not addressed or misinterpreted in our answers, please let us know and we will elaborate our response.
>
> Q1: Could the mean embedding be enough to capture the diversity of a large-scale (e.g. LAION, 400 million images) dataset?
>
> A1: In theory, an appropriate and sufficiently large embedding will be able to capture even highly complex and diverse data distributions if given enough samples, for instance when the features approximate a characteristic kernel as in DP-MERF. In practice however, the required embedding size and number of samples may be intractable. Given that the non-private version of DP-MEPF lacks behind the state of the art even on Cifar10, we don’t expect our VGG19-based embedding to accurately capture the diversity present in LAION or comparable datasets.
> This is acceptable to us because appropriate levels of DP prevent the modelling of highly diverse data anyway and so the limitations of DP-MEPF are overshadowed by those put in place through our privacy guarantee. By definition, DP prevents accurate learning of modes in the data distribution which are only represented by a handful of samples and this is in line with empirical findings (as e.g. with regards to fairness). Further, work by Feldman & Zhang [FZ2020] suggests that the well documented tendency of deep learning models to memorize datapoints actually plays a significant role in their ability to learn diverse distributions.
> The motivation of DP-MEPF is thus to accept that DP machine learning may not scale to the same complexity as its non-private counterpart, and instead aim to provide strong guarantees and useful results in the settings where it is possible. We would, however, like to emphasize that compared to prior methods in DP data generation, Cifar10 does constitute a large leap in data complexity and diversity.
>
> [FZ2020] Feldman, V., & Zhang, C. (2020). What neural networks memorize and why: Discovering the long tail via influence estimation. Advances in Neural Information Processing Systems, 33, 2881-2891.
>
>
> Q2: The authors could add some discussion on the assumption of a publicly available dataset. Following Tramer et al, 2022, we can question whether public datasets generally scraped from the web could be (1) considered as privacy-preserving and (2) available for certain tasks.
>
> A2: We include a discussion of the arguments of Tramer et al. in our broader impact statement. Please refer to the updated paper for the full text
>
>
> Q3: Moreover, acquiring information only through the DP channel is indeed more difficult but could alleviate these concerns (the reviewer does not ask for additional experiments but rather to give more substance to this discussion in the paper).
>
> A3: We design DP-MEPF explicitly to utilize the assumption of auxiliary public data and don’t require or expect it to perform well otherwise. To satisfy our curiosity, we experimented with splitting the privacy budget and using the first half to train a DP classifier and the second half to train DP-MEPF on the same data. We have added the results of this experiment to the appendix. The conclusion is, as expected, that DP-MEPF is not the method of choice if no public data is available.
>
> we continue in the next comment due to character limit...

---

> > ### Author Response · Authors · 2023-03-10
> > **(second half of our response)**
> >
> > Q4: The authors could mention recent/concurrent work on Differentially Private Stable Diffusion (Dockhorn et al, 2022)
> >
> > A4: The concurrent work by Dockhorn et al presents a major advance in generative modelling without public data. We will add it as a comparison. In the updated version published on openreview, the authors present results both on CelebA and Cifar10. The model (DPDM) achieves good results on CelebA, producing diverse images of faces with an FID score of 21.1 at ($\epsilon = 10, \delta = 10^{-6}$)-DP (still significantly worse than the DP-MEPF score of 11.7 in the same setting). However, in the higher privacy setting of ($\epsilon = 1, \delta = 10^{-6}$)-DP, DPDM only gets an FID of 71.8, while DP-MEPF maintains good performance with a score of 13.2. On Cifar10, DPDM struggles to produce recognizable images even in low privacy settings of ($\varepsilon = 10, \delta = 10^{-5}$)-DP, obtaining only an FID of 97.7. For reference, DP-MEPF has an FID of 26.6 in the same setting and still yields better scores at $\epsilon = 0.5$ with an FID of 64.4.
> > So the findings by Dockhorn et al also support our central argument that auxiliary public data, as used in DP-MEPF, significantly improves the ability to scale DP generative models to complex datasets.
> > DP diffusion models do apparently outperform DP-MEPF on several MNIST and FashionMNIST results. This may in part be explained by the significant difference between auxiliary data (SVHN and Cifar10) and private data (MNIST and FashionMNIST, respectively) in our experiments. Closer matches between auxiliary and private data would likely lead to better results, but since our focus lies on scaling to more complex data, we did not explore such alternatives in this setting. The comparison made by Dockhorn et al. should be taken with a grain of salt, as the exact downstream models used for reporting accuracy do not match the ones we have used, and thus some improvements may be due to better tuning of the classifiers, but we do not contest their claim that DP diffusion models perform better on MNIST, since it does not contradict the central message of our paper.
> >
> >
> > Q5: The authors motivate the need for synthetic image generation mainly through the angle of downstream tasks, for instance by generating private images that still maintain the classification accuracy of a given model. However, another angle is to be able to generate images from the original distribution that are private and consider these images as the end product. In that sense, a potential follow-up would be to perform empirical attacks on the generated images to see how close to the original training set these are.
> >
> > A5: The use-cases for DP generated data can be as diverse as for real data and as such, our evaluating of FID and downstream accuracy is by no means an exhaustive assessment of data quality.  We choose those two metrics because they are well-known and measure some of the common properties that we would like proxy data to have: it should look similar (FID) and behave similarly for one of the most common ML applications (downstream accuracy). If we understand your comment correctly, you propose trying to generate samples from the original distribution to a precision that they closely match individual training samples. We don’t believe this is desirable or attainable. For this to happen, the sample would either need to be memorised by the model, which DP is explicitly preventing, or the sample would need to be present several times in the dataset. The latter case would violate our implicit assumption that individual samples need to be protected, and instead likely require group privacy.

---

### Review · Reviewer_JBBh · 2023-02-21

**Summary Of Contributions:**

The paper proposes a new method for differentially private synthesis of data in the vision domain. The proposed method relies on existence of huge public training data, and differs from prior work by proposing a differentially private method that relies on MMD loss (minimizing the maximum mean discrepancy). The proposed method has three main steps, in the first step a feature extracted is trained on the public corpus, in the second step the mean embedding of the data distribution is computed and privatized (using the gaussian mechanism, and both first and second moment are used as features). Then, a generator which transforms noise to images is trained using the private features, with the MMD loss. There is also a private early stopping scheme that they suggest, which looks at a proxy FID score to determine when to stop training.

The authors then evaluate the proposed method on MNIST, FashionMNIST, CIFAR and CelebA datasets, using FID score and classification accuracy of downstream tasks.

**Audience:**

Yes

**Broader Impact Concerns:**

The paper does not have a broader impacts statement which is concerning, given the sensitivity of the issues covered.
My main concern in terms of broader impact is the fairness/disparity in drop of utility and representation of different subgroups in the synthesized samples. As shown by numerous prior work, differential privacy has disparate impact on different subgroups, especially a negative impact on underrepresented groups. This is particularly important in the case of this paper where the main goal is synthesis of data which is as similar as possible to the private data distribution, and one of the case studies is CelebA data, which is faces. In such a case, it is important to make sure different people from different subgroups are presented. I think a discussion/study of how the proposed method differs from normal DP-SGD+GAN training in terms of representation of different subgroups is needed.

**Claims And Evidence:**

Yes

**Requested Changes:**

To  address the mentioned weaknesses, I propose the following changes to help improve the manuscript:

1. I think at least a discussion of broader impacts is needed, with focus on the fairness of the method. I have elaborated more in the section regarding broader impacts, but to summarize, there needs to be a study where either the FID/downstream classification accuracy for each subgroup in the data, potentially based on race for the CelebA dataset, is measured for both private and non-private data synthesis, to see how badly the underrepresented races suffer from loss in utility, and how it is exacerbated using the proposed DP method, and how it compares with one of the baselines (either DP-SGD trained GAN or  DP-sinkhorn).

2. The results could be elaborated further/better, right now the progression is a bit confusing, since its based on dataset and not methods. Also, one very confusing thing is how the CelebA results are explained. Although I would assume Table 3 is a big part of the results, it is very briefly elaborated. Also, DP-sinkhorn is mentioned in the text for this dataset but there are no results in any tables for it, it is unclear where the FID 190 comes from or what the setup is there. I also have some doubts about the results in Table 3: why is there no 64*64 version of the DP-GAN? Also, the gap between DP-GAN and proposed method is a bit too high (FID 40 vs 11 for epsilon 10), how much tuning was done for the DP-GAN?

3. As mentioned above section 3 could be better explained. And one final minor issue, there is something odd about the formatting in page 10, beneath figure 7, the lines break at a strange place.

**Strengths And Weaknesses:**

Strengths:

1. At least to my knowledge and based on the claim in the paper, this is the first work in private data synthesis  for images that that uses auxiliary data and  provides meaningful synthesized samples of the CelebA dataset.

2. There are ample experiments to ablate different aspects of the proposed method, such as the early stopping. There is also Downstream task evaluation.


Weakness (further elaborated in the requested changes section):

1. There is no study of the disparate impact that the proposed DP method has on the different subgroups. This could be a good discussion for the broader impacts statement, which is also missing from the paper.

2. I think section three could be expanded a bit, to better explain the proposed method, specifically the discussion surrounding the generation of both labels and images together, I think that part needs to be clarified more, maybe even added to figure 1.

3. The results, although there is plenty of them, are not well elaborated, specially Table 3.

---

> ### Author Response · Authors · 2023-03-10
> **Thank you for your feedback!**
>
> Below, we answer your individual questions in detail or refer to the relevant section in the updated document. If one of your concerns is not addressed or misinterpreted in our answers, please let us know and we will elaborate our response.
>
> Q1: study FID/accuracy for subgroups in the data, potentially based on race for the CelebA dataset, measured for both private and non-private data synthesis, to see how badly the underrepresented races suffer from loss in utility, and how it is exacerbated using the proposed DP method, and how it compares with one of the baselines (either DP-SGD trained GAN or DP-sinkhorn).
>
> A1: The impact of DP on marginalised groups in the dataset is already an active area of research. Since the trade-off between DP and fairness is not specific to our approach, it is out of our scope of research to conduct additional experiments on this well-explored issue.
> We have added a broader impact statement to our paper where we include references to the ample literature on the subject and also provide some useful intuition on how the disadvantage of marginalised groups manifests in DP-MEPF, when applied to class-imbalanced data.
>
> Q2: One very confusing thing is how the CelebA results are explained. Although I would assume Table 3 is a big part of the results, it is very briefly elaborated.
>
> A2: We have elaborated our discussion. Please refer to the updated document passages highlighted in red.
>
> Q3: DP-sinkhorn is mentioned in the text for this dataset but there are no results in any tables for it, it is unclear where the FID 190 comes from or what the setup is there.
>
> A3: The FID score of 190 is the reported score from the paper by Cao et al., (table 3) with $(10, 10^{-6})$-DP for 32x32 resolution, and no auxiliary public data. The authors only provide results for this single setting and the score is significantly worse than ours (of course because this is not a fair comparison). We have added the score to Table 3 for completeness and indicated the source in the text.
>
> Q4: Table 3: why is there no 64*64 version of the DP-GAN?
>
> A4: We will add DP-GAN results for 64x64 resolution CelebA within the next few days and post an update.
>
> Q5: Also, the gap between DP-GAN and the proposed method is a bit too high (FID 40 vs 11 for epsilon 10), how much tuning was done for the DP-GAN?
>
> A5: We have conducted an exhaustive hyperparameter search for the DP-GAN baseline, as detailed in the appendix section B.1. We repeat the excerpt here for your convenience:
> The DP-GAN baseline for Cifar10 and CelebA uses the same generator as DP-MEPF with 3 residual blocks and a total of 8 convolutional layers and is paired with a ResNet9 discriminator which uses Groupnorm instead of Batchnorm to allow for per-sample gradient computation.
> We pre-train the model non-privately to convergence on downsampled imagenet in order to maintain the same resolution of $32 \times 32$ and then fine-tune the model for a smaller number of epochs. Results are the best scores of a grid search over the following parameters at $\epsilon=2$, which is then used in all settings: number of epochs $\{1, 10, 30, 50\}$ generator and discriminator learning rate separately for $10^{-k}$ and $3 \cdot 10^{-k}$ with $k \in \{3,4,5\}$, clip-norm $\{10^{-3}, 10^{-4}, 10^{-5}, 10^{-6}\}$, batch size $\{128, 256, 512\}$ and, as advised in \citet{dpgans_revisited}, number of discriminator updates per generator $\{1, 10, 30, 50\}$.
>
>
> Q6: I think section three could be expanded a bit, to better explain the proposed method, specifically the discussion surrounding the generation of both labels and images together. I think that part needs to be clarified more, maybe even added to figure 1.
>
> A6:  We have expanded the explanation of labelled data generation and added an equation detailing the construction of a labelled mean embedding. Please refer to the updated paper for details.
>
> Q7: A strange line break on page 10
>
> A7: Fixed, thanks.

---

> > ### Comment · Reviewer_JBBh · 2023-03-29
> > **updated paper**
> >
> > I viewed the updated version of the paper and it answers my concerns, especially on the broader impact. I thank the authors for addressing the concerns.

---

### Review · Reviewer_2VzY · 2023-02-24

**Summary Of Contributions:**

The paper proposes an interesting method to using public data for training a synthetic data generator with differential privacy guarantees. The method consists of a feature extractor (e.g. Resnet) trained on public data, and an estimator which computes empirical means and variances of the latent representations of the network. This estimator is evaluated on private data, and it is normalised and Gaussian noise is added and thus it is made DP. As a last step, a generator network is trained to generate synthetic data that is aimed to fit the internal representations of the aforementioned DP estimate that was evaluated on private data. By post-processing the resulting generator is DP with the same guarantees as the estimator (compositions -> DP accounting).

**Audience:**

Yes

**Broader Impact Concerns:**

no concerns.

**Claims And Evidence:**

Yes

**Requested Changes:**


Bigger questions I hope you can answer:

- You use early stopping based on the DP FID score estimator. Is early stopping rigorously DP? If you think not, could you mention that this part is heuristic, or if it is, could you shortly explain why it is? Using e.g. that Wang et al. (2019) analysis (RDP - analysis), I think early stopping is DP for sure for the epsilons and deltas corresponding to the max number of iterations, but if you compute the epsilons using the number of iterations up to the early stopping, then I am unsure.

- Could you clarify: how are the functions $\phi_1$ and $\phi_2$ normalized? Do you clip or just normalise them to 1, by enforcing or by implementing this in the network activations such that it happens automatically?

- Why is the sensitivity of the concatenated vector $[ \phi_1, \phi_2]$  2?  Isn't it $\sqrt{2}$ if you use 2-norm? It seems that in most experiments the alternative that uses $[ \phi_1, \phi_2]$ is worse than the one that uses only $\phi_1$, perhaps this might be due to the utility loss you incur here?

- Or if there is another reason, for the " $[ \phi_1, \phi_2]$ - alternative " being worse in the experiments, could you clarify?



Small things:


- p.3: The notation $\Phi$ for the concatenated features (values and their squares) does not seem to be used later (you could remove it I guess).

- Perhaps change the following sentence somehow, it felt a bit out of place in the beginning: " However, one of the properties of DP is composability, meaning data can be accessed more than once – but the level of privacy guarantee degrades each time. "

- p. 9: " achieves and FID"-> "an FID"

- p. 10: something strange with a linebreak

**Strengths And Weaknesses:**

Stenghts:

- The idea seems interesting and novel and widely applicable, the feature extractor can be any network as well as the generator.

- Strong experimental results. The lack of existing baselines for DP generative models that would exploit synthetic data underlines the novelty of this work.

- Theoretical guarantees that illustrate the utility preservation of the MMD estimator (obtained from squared norm of the difference of the estimator with synthetic data and the private estimator). The results are qualitative but seem to illustrate well how things scale w.r.t. noise level and number of private data points etc. Suggestion: perhaps as a next step here could be e.g. sigmas and dataset sizes replaced by deltas and epsilons.

Weaknesses:

- Mainly small things, mentioned below in "Requested Changes". I hope you can answer my question regarding the early stopping.

---

> ### Author Response · Authors · 2023-03-10
> **Thank you for your feedback!**
>
> Below, we answer your individual questions in detail or refer to the relevant section in the updated document. If one of your concerns is not addressed or misinterpreted in our answers, please let us know and we will elaborate our response.
>
> Q1: Explain why early stopping is DP
>
> A1: We release both the data embedding used for generator training (e.g. from VGG19) and the embedding used for early stopping (from the Inception network) exactly once at the beginning of training. These are already privatised so, we simply re-use these without incurring the privacy loss during the training. Our interpretation of your question suggests a misunderstanding of how often the latter embedding is released. If a new release were made every time we checked the early stopping condition, we would indeed have a dilemma of how to allocate the privacy budget, not knowing the total number of releases in advance, but this is not the case here. We hope this answers your question and will be happy to elaborate if you have additional questions or if we missed your point.
>
> Q2: Clarify how the functions $\phi_1$ and $\phi_2$ are normalised.
>
> A2: The procedure for normalising $\phi_1$ and $\phi_2$ is identical. Per sample in the dataset, the respective feature activations are extracted and concatenated into a vector, which is $\phi_1$. In the case of $\phi_2$, each entry in the vector is then squared. Afterwards, each vector is divided by its norm, scaling it to have norm 1. The rescaled vectors of all samples are then averaged.
>
> Q3: What is the sensitivity of the concatenated vector? $2$ or $\sqrt{2}$?
>
> A3: the per-sample embeddings $\phi_1$ and $\phi_2$ are separately released with the Gaussian mechanism, which is why the sensitivity of the concatenated vector is not mentioned. Each of the two embeddings is normalised to L2-norm 1. Since we use bounded DP (i.e. neighbouring datasets differ by exchanging one sample for another) and the maximum L2-distance between two sample-embeddings is 2, the sensitivity of the mean embedding is 2/m.
>
>
> Q4: the $(\phi_1,\phi_2)$ embedding seems worse in most settings. Why is that?
>
> A4:  The privacy-utility trade-off is different for the $(\phi_2,\phi_2)$ embedding. By doubling the dimension of the embedding, the DP release requires more noise. If the additional information provided by the second moment embedding outweighs the loss of information due to increased noise, the $(\phi_1,\phi_2)$ embedding should yield better results than the $\phi_1$ embedding. Based on our results, adding the second moment embedding $\phi_2$ is useful on MNIST, FashionMNIST and in the majority of experiments on Cifar10.
>
> Q5: page 3: The notation $\Phi(x)$ for the concatenated features (values and their squares) does not seem to be used later (you could remove it I guess).
>
> A5: The notation $\Phi$ is used throughout the proofs to refer to the full feature map.
>
> Q6: Perhaps change the following sentence somehow, it felt a bit out of place in the beginning: " However, one of the properties of DP is composability, meaning data can be accessed more than once – but the level of privacy guarantee degrades each time. "
>
> A6: Thanks for pointing it out. We have rephrased the sentence (see updated document for context): “DP even offers a means of tracking the effect of multiple accesses to the same data on its overall privacy level, but with each access, the privacy of the data gradually degrades.”
>
> Q7: page 9: " achieves and FID"-> "an FID"
>
> A7: Fixed, thanks.
>
> Q8: A strange line break on page 10
>
> A8: Fixed, thanks.

---

> > ### Comment · Reviewer_2VzY · 2023-03-10
> > **reply**
> >
> > Thank you for the answers! The answer related to early stopping clarifies things.
> >
> > About the L2-sensitivity: that also clarifies that you use the substitute neighboring relation. I would still say, that in your equation 5, you would get a bit 'cleaner' solution if you would consider the release of both $\mu_{\phi_1}$ and $\mu_{\phi_2}$ as a single Gaussian mechanism, with sensitivity $\sqrt{2^2 + 2^2} = \sqrt{8}$. The bound for that is the same as optimal bound for the composition of two Gaussian mechanisms with sensitivities 2 each, however you can simply use the 'analytical Gauss' formula by Balle and Wang (2018).

---

> > > ### Author Response · Authors · 2023-03-10
> > > **single Gaussian mechanism**
> > >
> > > yes, this is true! There will be a gain in using a single Gaussian mechanism than two, as we did. These image datasets mostly have many data samples, enough to have a good privacy-accuracy trade-off in our method. So, we didn't think hard about using one Gaussian than two. But we agree that for small datasets, using what the reviewer suggests makes sense.

---

> > > > ### Comment · Reviewer_2VzY · 2023-03-13
> > > > **reply**
> > > >
> > > > One more short remark: the dot after the equation of Proposition 4.1 seems to be off in the revised version.

---

### Author Response · Authors · 2023-03-10
**Uploaded revised paper and individual responses to reviewers**

We thank all the reviewers for their effort in carefully reading our manuscript and sharing their constructive feedback. We have updated our paper accordingly with the changes marked in red and respond to the questions raised by each reviewer individually below.

Results for the DP-GAN CelebA baseline at $64 \times 64$ resolution will be added within the next few days. The document will then be updated again.

---

> ### Author Response · Authors · 2023-03-16
> **DP-GAN CelebA 64x64 results added**
>
> We have now added DP-GAN results on CelebA 64x64 to Tables 3 and 9 as well as Figure 3. While the FID scores are a bit better than for the 32x32 resolution version, they are still significantly outperformed by DP-MEPF.
>
> We have also corrected a formatting error in Proposition 4.1.

---

### Author Response · Authors · 2023-07-17
**Paper revision due to error in FID evaluation**

After publication of the camera ready version of this paper, we discovered that the published experiments contained an error in the FID evaluation, where samples were incorrectly normalized when passed to the pytorch-FID package. This led to faulty and overly optimistic FID scores in Tables 3 - 5 in the main paper and Tables  7, 9, and 12 - 19 in the appendix.

We have now corrected our error by retraining all affected models and scaling the generated data to the correct [0,1] range when computing FID. The resulting scores are around 20% - 30% higher both for our method DP-MEPF and for our baseline of a pre-trained DP-GAN. Due to the advantage of using auxiliary public data, our approach outperformed other comparison methods by a substantial margin and our new results continue to support this finding. For instance, on CelebA 32x32 at $\varepsilon = 10$, we now obtain an FID score of 16.3 for DP-MEPF, which outperforms DP Diffusion Models (Dockhorn et al 2022) at 21.2, DP-Sinkhorn 189.5 and our pre-trained GAN at 58.1. At $\varepsilon = 1$, DP-MEPF achieves a score of 17.2, while DP Diffusion models (without public data) has a much higher score of 71.8.

With these results, our experiments continue to support the central claim of our paper: DP-MEPF can utilize auxiliary public data to generate differentially private image datasets of higher complexity and at higher quality than prior methods which do not make use of public data, and outperforms the baseline apporach of fine-tuning a pre-trained GAN.

We have submitted a revised version of the paper for consideration to the action editors and editors in chief. For the full list of updated results, please refer to this version, once it has been approved and uploaded.

---

> ### Comment · Action_Editors · 2023-07-19
> **Revision updated**
>
> After inspecting the revised version with the corrected FID evaluation, I confirm that the results still support the claims made in the original accepted version. Specifically, while the error scores for the proposed methods are a bit higher, this is also the case for some of the baselines (which were also relying on the incorrect scaling), and the performance of the proposed approach remains significantly better than competitors in the regimes of interest.
>
> The revised version of the paper has been uploaded.
>
> Best,
> The AC

---

### Decision · Action_Editors · 2023-04-14

**Recommendation:** Accept as is

**Comment:**

The authors made convincing answers to the reviewers' questions and revised their manuscript accordingly. After the discussion phase, there is a consensus on accepting the paper.

**Audience:**

The topic of private data generation is of wide interest to the ML community.

**Claims And Evidence:**

This paper proposes a new approach for DP image generation based on (1) pretraining features on public auxiliary data, and (2) minimizing a MMD term with respect to the privatized mean embedding. The approach is claimed to achieve superior privacy-utility trade-off as previous work that does not use auxiliary data. This claim is matched by convincing empirical evidence.